



# An *in situ* gas chromatograph with automatic detector switching between Vocus PTR-TOF-MS and EI-TOF-MS: Isomer resolved measurements of indoor air

Megan S. Claflin[1], Demetrios Pagonis[2,3], Zachary Finewax[2,3,4], Anne V. Handschy[2,3], Douglas A. Day[2,3],
Wyatt L. Brown[2,3], John T. Jayne[1], Douglas R. Worsnop[1], Jose L. Jimenez[2,3], Paul J. Ziemann[2,3],
Joost de Gouw[2,3], Brian M. Lerner[1]

[1] Aerodyne Research Inc., Billerica, Massachusetts 01821, United States
[2] Cooperative Institute for Research in Environmental Sciences (CIRES), Boulder, Colorado 80309, United States
[3] Department of Chemistry, University of Colorado, Boulder, Colorado 80309, United States
[4] Now at Chemical Sciences Laboratory, Earth System Research Laboratory, National Oceanic and Atmospheric Administration, Boulder, Colorado, 80305, United States

*Correspondence to*: Megan S. Claflin (mclaflin@aerodyne.com)

**Abstract.** We have developed a field-deployable gas chromatograph (GC) with thermal desorption preconcentration (TDPC), which is demonstrated here with automatic detector switching between two high-resolution time-of-flight mass spectrometers (TOF-MS) for in situ measurements of volatile organic compounds (VOCs). This system provides many analytical advances including acquisition of fast time-response data in tandem with molecular speciation and two types of mass spectral information for each resolved GC peak: molecular ion identification from Vocus proton transfer reaction (PTR) TOF-MS and fragmentation pattern from electron ionization (EI) TOF-MS detection. This system was deployed during the 2018 ATHLETIC campaign at the University of Colorado Dal Ward Athletic Center in Boulder, Colorado where it was used to characterize VOC emissions in the indoor environment. The addition of the TDPC-GC increased the Vocus sensitivity by a factor of 50 due to preconcentration over a 6 min GC sample time versus direct air sampling with the Vocus which was operated with a time resolution of 1 Hz. The GC-TOF methods demonstrated average limits of detection of 1.6 ppt across a range of monoterpenes and aromatics. Here, we describe the method to use the two-detector system to conclusively identify a range of VOCs including hydrocarbons, oxygenates and halocarbons, along with detailed results including the quantification of anthropogenic monoterpenes, where limonene accounted for 47 – 80% of the indoor monoterpene composition. We also report the detection of dimethylsilanediol (DMSD), an organosiloxane degradation product, which was observed with dynamic temporal behavior distinct from volatile organosiloxanes (e.g. decamethylcyclopentasiloxane, D5 siloxane). Our results suggest DMSD is produced from humidity-dependent, heterogeneous reactions occurring on surfaces in the indoor environment, rather than formed through gas-phase oxidation of volatile siloxanes.


## 1 Introduction

Historically, volatile organic compound (VOC) emissions from transportation were the most important air pollution source in urban environments (Gentner et al., 2017; Watson et al., 2001). However, with the success of emission reduction strategies (Warneke et al., 2012; McDonald et al., 2013), other sources of anthropogenic VOCs are becoming significant in most developed nations, such as emissions from volatile chemical products (VCPs) (McDonald et al., 2018). VCPs consist of a large diversity of compounds including oxygenated species like alcohols (e.g. glycols), esters, siloxanes, and carbonyls along with hydrocarbons like alkanes, alkenes (e.g. monoterpenes), and aromatics (McDonald et al., 2018). This emission class stems from human activities such as the use of personal care products, paints, cleaning supplies, pesticide application, and the industrial use of solvents. Typically, VCPs are emitted in residential or commercial buildings, making their emissions highly variable both spatially and temporally depending on the occupancy and activities occurring in the space (Weschler and Carslaw, 2018; Abbatt and Wang, 2020; Pagonis et al., 2019). To understand changing emission patterns, analytical instrumentation that can quantitively detect these classes of VOCs with little ambiguity and high time resolution is needed along with a range of studies to understand how emissions differ depending on the indoor environment and its use.

While indoor air quality has been studied for decades (Weschler and Shields, 1997; Wolkoff, 2013), recently the use of advanced gas-phase analysis techniques developed for atmospheric research, like in situ (real-time, direct air sampling) proton transfer reaction (PTR) and chemical ionization (CI) mass spectrometry (MS), have been applied for the characterization of indoor VOCs. These techniques have been used to characterize emissions in indoor environments such as a movie theater (Williams et al., 2016), art museum (Pagonis et al., 2019; Price et al., 2019), university classrooms (Liu et al., 2016, 2017; Tang et al., 2015, 2016), and to study how episodic events like cleaning and cooking impact indoor air quality (Wong et al., 2017; Kristensen et al., 2019; Lunderberg et al., 2019). While these studies conducted with PTR-MS and CIMS provide VOC emission signatures in a variety of environments, they often cannot provide molecular identification due to the detection of isobaric ions, which can be associated with multiple isomers, cluster ions, or fragmentation products that have the same molecular formula (Thompson et al., 2017). Without molecular identification, source apportionment and fate characterization remain difficult.

Improved molecular information can be gained by coupling gas chromatography (GC) with mass spectrometric detection (Warneke et al., 2003). Some studies have conducted off-line GC measurements for indoor air research, which generally consist of sorbent tube or solid phase microextraction (SPME) fiber collection with subsequent GC analysis (Gallagher et al., 2008; He et al., 2019; Sun et al., 2017; Liu et al., 2019). These studies have focused on emissions from human skin and breath (Gallagher et al., 2008; He et al., 2019; Sun et al., 2017), with the exception of Liu et al. (2019), which utilized off-line GCxGC analysis to study VOCs in a single-family home in northern California.

While these approaches provide some molecular identification and quantification, the low time resolution and time-consuming nature of off-line methods along with the potential for the introduction of artifacts due to sample handling between collection



and analysis are not ideal. In situ GC measurements of indoor environments are currently limited (Kristensen et al., 2019; Lunderberg et al., 2019; Rizk et al., 2018). During the single-family house study mentioned above (Liu et al., 2019), a semi-volatile thermal desorption aerosol gas chromatograph (SV-TAG) was deployed to make measurements during normal occupancy (Kristensen et al., 2019; Lunderberg et al., 2019). In the summer of 2018, an intensive indoor air study, HOMEChem, was conducted to study emissions and removal processes of gases and particles in a model home. This campaign included SV-TAG, an in situ 4-channel GC with flame ionization detection (FID) and electron capture detection (ECD), and passive sampling for off-line GC-MS samples (Farmer et al., 2019). The use of multiple types of chromatographic separation during this campaign illustrates the shift in focus for indoor air research toward more complete molecular analysis.

Building upon the research that has been conducted to study indoor environments, the ATHLETic center study of Indoor Chemistry (ATHLETIC) campaign was conducted during November of 2018 at the University of Colorado Dal Ward Athletic Center in Boulder, Colorado. The goal of ATHLETIC was to quantify the effects of human exercise, the use of chlorine-based cleaners, and other parameters on indoor air quality with instrumentation that provides high-time resolution information and detailed characterization of both gases and particles. To address the need for high time resolution measurements and molecular identification of VOCs we have developed an automated, field-deployable GC equipped with thermal desorption (TD) preconcentration and automated detector switching between two high-resolution time-of-flight mass spectrometers (HR-TOF-MS): a Vocus PTR-TOF-MS and an electron ionization (EI) TOF-MS for in situ measurements of VOCs. This system was deployed during the 2018 ATHETIC campaign to characterize VOC emission profiles in the weight room facility. The instrument configuration and details of operation are discussed here along with measurement results that were made possible through the analytical advances this technique offers. These results include the identification of a range of VOCs including hydrocarbons, oxygenates, and halocarbons in the athletic center along with details of their detection by both types of TOF-MS. We also report the quantification of anthropogenic monoterpenes and evidence of VOC emissions from humidity-dependent, heterogeneous reactions occurring on walls and surfaces in the indoor environment. The results presented here are a demonstration of this new GC-TOF-MS technique that produces three detailed and complementary data sets.

## 2 Methods

### 2.1 Instrument Description

The GC-TOF-MS system consists of three main components: (1) thermal desorption pre-concentrator (TDPC) for sample collection, (2) gas chromatograph (GC) for sample separation, and (3) high resolution time-of-flight mass spectrometers (HR-TOF-MS) for sample detection. Each of these components is described in the following sections. While the *in situ* GC can be operated with either the Vocus PTR-TOF-MS or EI-TOF-MS as individual detectors, coupling the GC with both detectors creates a technique that produces three complementary data sets: (1) real-time Vocus PTR-TOF-MS, (2) GC-Vocus PTR-TOF-MS, and (3) GC-EI-TOF-MS. Hereafter, these three techniques will be referred to as RT-Vocus, GC-Vocus, and GC-EI-TOF,



respectively. It should be noted that the instrument described in this work, and deployed for the ATHLETIC campaign, was a

prototype system used to demonstrate this technique. The instrument is undergoing continued development to improve sensitivities, chromatographic performance, and extend the volatility range of resolved compounds since this campaign.

## 2.2 Thermal Desorption Preconcentration

Ambient VOCs are typically present in low mixing ratios (sub-ppb), and so to increase GC-MS sensitivity a preconcentration method is required. For this study, the samples were collected using a simplified version of a thermal desorption

preconcentrator (TDPC) (Aerodyne Research, Inc.). Briefly, the TDPC employed for this study relied upon a single-stage adsorbent trap for pre-concentration of analytes. The design is based upon that of Tanner et al. (2006), and uses a commercial cold-plate Peltier thermoelectric cooler (CP-110, TE Technology) to allow for precise ambient to sub-ambient temperature regulation. Results from this TDPC have been described previously (Anderson et al., 2019). The system was simplified by not using water trapping or oxidant scrubbing before sample collection due to the expected low humidity and oxidant mixing ratios

in this study. The sample trap was a commercial glass sorbent tube (TO-15/TO-17 cold trap, Markes International) operated at 20°C during sample collection to avoid potential water condensation. The chosen sample trap was a multi-bed adsorbent trap equipped with 3 stages of adsorbents (Tenax, Carbopack X, Carboxen 1003; personal comm., Markes International, 2020) to expand the volatility range of compounds that can be trapped and desorbed for analysis. The combination of adsorbents in the TO-15/TO-17 trap allows for the analysis of a wide range of VOCs (including oxygenates) in the $C_2 - C_{32}$ $n$-alkane volatility

range. Details of operational parameters (e.g. temperatures, flows) are described in Section 2.8.

## 2.3 Gas Chromatograph

To separate analytes before detection with TOF-MS, a compact GC from Aerodyne Research, Inc. (hereafter referred to as ARI GC) was used. The ARI GC is designed to be an *in situ*, field-deployable system. It fits into a 55 cm x 55 cm x 30 cm rack, weighs 24 kg, consumes 300 W of power during typical operation, and contains all hardware for GC sample collection

and control of TDPC and GC flows and temperatures, including a make-up flow needed for GC-Vocus measurements (described in Section 2.8). Here, the flow path contained three 2-position chromatography valves with nitronic 60 valve bodies (VICI Instruments): one 10-port and two 6-port valves (Figure 1B) to direct flows during the GC cycle. The chromatography valves and transfer lines (Sulfinert-treated 304-SS, 1.6 mm OD, 0.76 mm ID, Restek) are housed in a heated enclosure held at 150°C. The carrier gas (UHP helium; Matheson) was controlled by a mass flow controller (MKS Technology) with variable

setpoint capability in the range of 0.1 – 10 cm³ min⁻¹. The GC column is housed in a custom interlocking aluminum spindle (12 cm x 3 cm) with surface-mounted flexible resistive heaters, as described by Lerner et al. (2017). For this study, the ARI GC was configured as a one-channel system (single column separation), with a 30 m Rxi-624 analytical column (Restek, 0.25 mm ID, 1.4 μm film thickness) installed in the spindle. This column resolves non- to mid-polarity VOCs including hydrocarbons, oxygenates, and nitrogen and sulfur containing compounds, with the exception of high polarity compounds like





carboxylic acids. The volatility range that the GC can resolve is a function of both the chosen GC column and the TDPC

adsorbent trap. With the combination of column and adsorbent trap used for this study, the ARI GC was optimized for $C_5 - C_{12}$ hydrocarbons, along with oxygen, nitrogen, halogen, and sulfur containing VOCs.

## 2.4 HR-TOF-MS Detection

### 2.4.1 EI-TOF-MS

The electron ionization mass spectrometer used in this study is a Tofwerk EI-TOF-MS (Tofwerk AG) that has been described

previously (Obersteiner et al., 2016). While the EI-TOF has nominal mass resolution up to 5000 $m/\Delta m$, here it was operated

with a resolution of 3900 at $m/z$ 69 to optimize both mass resolution and instrument sensitivity. During acquisition, mass

spectra were averaged on a 6 Hz time base to obtain enough data points across each chromatographic peak. The ionizer

temperature was kept at 280°C, with ionization energy set to 70 eV and an electron emission current of 0.3 mA. The interface

between the GC and both EI-TOF and Vocus is described in Section 2.8.

### 2.4.2 Vocus PTR-TOF-MS

The proton transfer reaction mass spectrometer used in this study is a Tofwerk Vocus PTR-TOF-MS (Tofwerk AG) described

by Krechmer et al. (2018). It has nominal resolution of 12000 $m/\Delta m$ and was operated with a resolution of 11500 at $m/z$ 150.

The Vocus was operated with a data acquisition rate of 1 Hz for RT-Vocus and 5 Hz for GC-Vocus measurements. The

focusing ion-molecule reactor pressure was maintained at 1.5 mbar, giving a reduced electrical field ($E/N$) of 150 Td.

Additional details of Vocus operation during this campaign are given in Finewax et al. (2020).

## 2.5 Instrument Control, Data Acquisition and Analysis

ARI GC operation is fully automated via a Labview-based (National Instruments, Inc) stand-alone executable in a Windows

10 OS environment (Microsoft) on one of the TOF-MS computers (here, the EI-TOF computer was used). The ARI GC

communicates with the control computer via USB 2.0, with two communication devices (data board, serial communications

board) required for operation. Each mass spectrometer is equipped with its own acquisition software (Tofwerk AG), the EI-

TOFMS operating TofDAQ v.1.99 and the Vocus using Igor Pro-based (Wavemetrics) Acquility v.2.3.6 which acts as a

command shell and GUI interface for TofDAQ.

The analysis of high-resolution mass spectrometric data from both the EI-TOF and the Vocus was performed using Tofware

(v3.1.2; TofWerk AG and Aerodyne Research, Inc.), where both nominal (unit mass resolution, UMR) and accurate (high-

resolution, HR) data were used for analysis. Once the data had undergone mass calibration and high-resolution ion peak fitting,

the data was then imported into GC analysis software, TERN (Aerodyne Research, Inc.). TERN is a software package, based

in Wavemetrics Igor Pro that automatically calculates chromatographic peak areas, for either UMR or HR data, by





mathematically fitting peak functions to the data rather than peak integration (Isaacman-VanWertz et al., 2017). Instrument
calibration and data normalization procedures employed for this study are described in Section 2.9.

## 2.6 Measurement Site

ATHLETIC was a three-week study conducted at the University of Colorado Dal Ward Athletic Center in November 2018 in
Boulder, Colorado. During the campaign, instruments were housed within the athletes' weight room and sampled from both
inside the weight room (hereafter "room air") and the supply air from the heating, ventilation, and air conditioning (HVAC)
system. During the measurement period, the instruments switched between sampling the room and supply air every 10 min via
an automated valve system. The weight room is serviced by the main air handling unit (AHU) of the building that circulates
~400–1400 $m^3$ $min^{-1}$, of which 200 $m^3$ $min^{-1}$ is supplied to the weight room. The fraction of outside air that was mixed with
the main AHU flow varied from ~10–80% during this study. The volume of the weight room is ~1700 $m^3$, which corresponds
to an average residence time of air in the weight room of ~8.5 min, and an outdoor air exchange rate of 0.7 – 5.6 air changes
per hour (ACH). The Dal Ward Athletic Center is directly adjacent to the University of Colorado football stadium, Folsom
Field. The athletic center is to the north of the football stadium and to the northeast of a fieldhouse where cooking and other
activities occurred before and during two football games that took place during this study on November 10 and 17.

The campaign included additional instrumentation that sampled gases and particles. Although ATHLETIC was a three-week
study, the GC-TOF-MS system operated for a subset of the campaign. Here, only results from the GC-TOF-MS system will
be presented along with relative humidity (RH) and temperature data collected using a Picarro Gas Analyzer (G2401) and
building space temperature sensors located on the walls of the main floor of the weight room (provided and operated by CU
Facilities Management). Room RH was derived from the building temperature and local pressure along with the $H_2O$ mixing
ratio measured by the Picarro instrument. A separate analysis of RT-Vocus data, focusing on species not discussed here, is
published elsewhere (Finewax et al., 2020).

## 2.7 Sample Inlet

The ARI GC houses three separate sample inlets, an ambient inlet and two calibration gas inlets (Figure 1B). The GC ambient
inlet sampled from the weight room via a 3.4 m PFA (0.16 cm OD) sampling line with 30 $cm^3$ $min^{-1}$ flow rate. The two
calibration gas inlets are for pressurized gases where each inlet has a critical orifice inline to regulate flow followed by a
solenoid shut-off valve. The calibration inlets operate by overflowing the ambient inlet during the sampling period; this excess
flow is ensured by setting the pressure on the gas cylinder regulator based upon the critical orifice diameter (typical size 75
µm) installed upstream of the solenoid valves. For this study, the calibration gases were (1) a custom-made multicomponent
calibration mixture: a certified natural gas standard (Restek) diluted with UHP nitrogen and (2) a zero gas (ultra zero grade
air, Airgas) for system blanks.



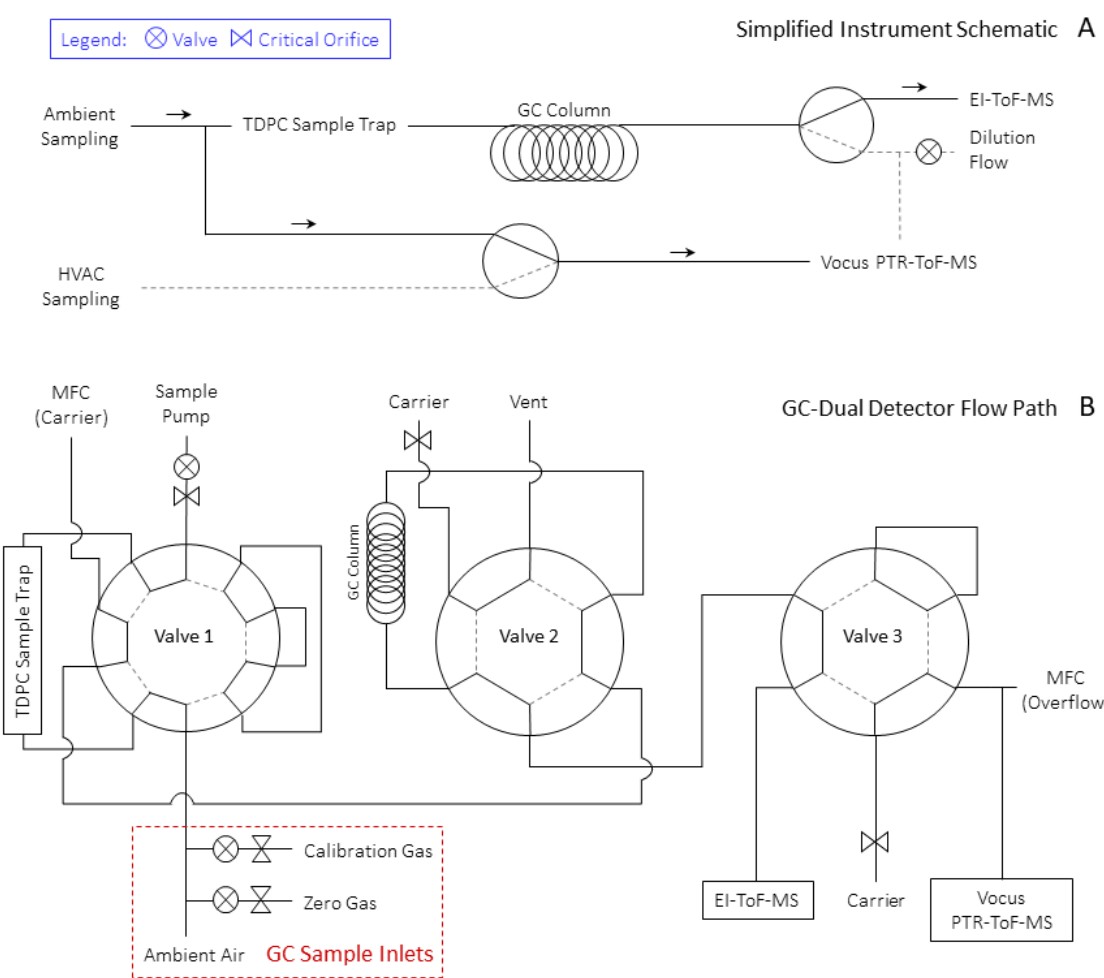

**Figure 1. Instrument schematics of (A) dual detector GC-TOF-MS instrument configuration with valving shown for GC detector selection (EI-TOF or Vocus) and Vocus inlet source (room or supply air or GC effluent) (B) GC flow path and valve positions to incorporate a single stage thermal desorption preconcentrator (TDPC), single column separation, and dual TOF-MS detection.**

For RT-Vocus sampling, room air was sampled at 10 L min$^{-1}$ through a 1.3 m length PFA Teflon inlet with 0.47 cm inner
diameter (ID) that was shared by all instruments. Supply air was sampled at the same flow rate through a 4.3 m length of PFA
Teflon with the same ID. From those shared inlets, 1.6 L min$^{-1}$ was pulled through a 1.5 m PFA (0.16 cm ID) sampling line,
where 100 cm$^3$ min$^{-1}$ was sampled into the Vocus and the remainder to excess. Sample selection (room versus supply air) was
done via automated valve switching, and a makeup flow was applied to the inlet not being sampled to ensure continued inlet
passivation. The RT-Vocus room air inlet and GC ambient inlet were separate but co-located in the weight room. The GC did
not sample from the supply air during this study.



## 2.8 Sample Acquisition, Separation and Detection

At the start of the GC cycle (22 min), the sample was collected onto the adsorbent trap held at 20 +/- 1°C for 6 min at 30 $cm^3$ $min^{-1}$. The adsorbent trap was then backflushed for 1 min with 2 $cm^3$ $min^{-1}$ of UHP helium (Matheson) to remove oxygen and water from the trap. Next, the carrier flow was increased to 5 $cm^3$ $min^{-1}$ and the sample was thermally desorbed onto the GC column by flash heating the adsorbent trap to 225°C for 20 s at 10.5 °C $s^{-1}$. During this sample transfer, the GC column was held at 40°C.

The method of chromatographic separation was as follows: after sample transfer the column temperature was held at 40°C for 40 s, then ramped from 40°C to 100°C at 40°C $min^{-1}$, increased to 150°C at 15°C $min^{-1}$, and finally increased to 225°C at 30°C $min^{-1}$ and then held at 225°C for 100 s for a total chromatogram time of 9 min. Following the separation, the GC column was backflushed with UHP helium for 140 s while held at 235°C and the sample trap was switched out of the path and backflushed with 2 $cm^3$ $min^{-1}$ UHP He while heated to 225°C for 20 s to prepare the adsorbent trap for the next sample collection. After this cleaning period, the column was cooled and held at 40°C over 575 s with continuous backflushing until the next sample transfer. Column flow rate was 2 $cm^3$ $min^{-1}$ of UHP helium controlled by a mass flow controller during sample transfer and chromatography, and by a critical orifice during backflushing.

After the chromatographic separation, the column effluent was directed to either the EI-TOF or the Vocus for detection. The ARI GC was coupled to the EI-TOF by a fused-silica capillary line (Siltek guard column, 0.25 mm ID, Restek) which passed through a heated capillary feedthrough, kept at 230°C, so that the GC effluent was directed into the ionization region of the EI-TOF. For GC-Vocus measurements, the ARI GC was coupled to the Vocus by a passivated stainless steel (Sulfinert treated, 1.6 mm OD, 0.76 mm ID, Restek) transfer line heated to 150°C. For GC-Vocus measurements, the total flow provided from the GC must be adequate for the Vocus ambient pressure inlet, which is fixed at 100 $cm^3$ $min^{-1}$ via a PEEK capillary tube and pressure controller. Since the GC column effluent flow rate is 2 $cm^3$ $min^{-1}$, an additional make-up flow of 100 $cm^3$ $min^{-1}$ of zero grade air (Airgas) was introduced upstream of the GC-Vocus transfer line, downstream of the GC column (Figure 1B).

## 2.9 Normalization of Instrument Response and Calibration

Normalization of the EI-TOF data is required to account for changes in instrument sensitivity due to changes in detector response. For this study a normalization method that utilizes the detection of long-lived halocarbons in the atmosphere was used, as described previously by Lerner et al. (2017). Specifically, ambient carbon tetrachloride ($CCl_4$) was used here to normalize the EI-TOF data, as no sources were expected in the weight room, consistent with a lack of significant variations in its time series. A normalization factor (NF) was calculated for each sample by dividing the GC peak area of $CCl_4$ by the average $CCl_4$ peak area for the entire campaign. The EI-TOF data was then corrected by dividing the raw data by the NF.

The Vocus signal is dependent on the concentration of the analyte, time spent in the ion-molecule reactor, the rate of reaction, and the concentration of the reagent ion (Yuan et al., 2017). Variability of analyte signal is reduced by normalizing to a constant





reagent ion signal of 1 x 10$^6$ counts per second (cps). During the ATHLETIC campaign, the largest signal observed in the Vocus was $(H_2O)_3H^+$; actual ion concentrations in the reactor were dominated by $H_3O^+$ ions, but these are poorly measured due to mass discrimination in the quadrupole ion guide between the reactor and time-of-flight analyzer. Analyte signal was
divided by $(H_2O)_3H^+$ and multiplied by 1 x 10$^6$ cps to obtain normalized signal. Monoterpene calibration of the RT-Vocus signal was accomplished immediately following the campaign, where a 6-point calibration of a gravimetric standard of limonene was diluted in zero air, resulting in a calibration factor of 139 ncps/ppbv for $C_{10}H_{17}^+$. The validity of using a single monoterpene to generate a general sensitivity for the RT-Vocus to represent the mixture of monoterpenes present in ambient air will be discussed in Section 3.3.1. Backgrounds for the Vocus were obtained using a zero-air generator for 30 s every 15
min, and subtracted with linear interpolation between background collection periods (Krechmer et al., 2018).

At the end of the campaign both the GC-EI-TOF and GC-Vocus were calibrated for aromatic compounds and monoterpenes. This was accomplished by performing a 3-point calibration curve with duplicates for each detector using a 3 L Tedlar bag (Restek) containing 2 ppb each of BTEX (benzene, toluene, ethyl benzene, m-, p-, and o-xylene) and five monoterpenes. The calibration sample was made by diluting a standard solution of BTEX (Restek) with methanol (HPLC grade, Sigma-Aldrich),
and creating a solution of five standard monoterpenes (α-pinene (99%), β-pinene (99%), camphene (96%), carene (97%), and limonene (99%) (Sigma-Aldrich)) by diluting in hexanes (HPLC grade, Sigma-Aldrich). Using a glass microsyringe, 5 μL of each of these solutions was then injected into a stream of UPH $N_2$ (Airgas) flowing into the Tedlar bag at nominally 500 cm$^3$ min$^{-1}$. This mixture was attached to the sample inlet of the GC for calibrations. For the GC calibrations the sampling rate of the GC was kept constant (30 cm$^3$ min$^{-1}$) but the collection time was varied (1 min, 3 min, and 6 min of collection) to generate
the calibration curves. This created a curve that gave instrument response (normalized counts; ncts) versus sample volume. By dividing the response at each sample volume by the compound concentration in the Tedlar bag, the instrument sensitivity (ncts ppb$^{-1}$) was calculated. The difference in units between the RT-Vocus and the GC data should be noted; the RT-Vocus data is reported as normalized counts per second (ncps), while the GC data is given as normalized counts (ncts), the integration of detector response across the peak elution time. Figure S1 shows calibration curves of GC-TOF instrument sensitivity versus
sample volume of selected monoterpenes. The limits of detection (LOD) were calculated as 3 times the standard deviation of the baseline multiplied by the full width half maximum (FWHM) of the chromatographic peak, then divided by the sensitivity. Calibration results for both the GC-EI-TOF and GC-Vocus, including ions used for quantification, instrument sensitivity, LODs, and correlation coefficients ($R^2$), are given in Table 1. Available RT-Vocus sensitivities measured for this campaign are also reported in Table 1 for comparison with the GC sensitivities. The ambient GC data was converted to mixing ratio by
dividing the normalized signal with the sensitivity.



**Table 1. Quantification ions, measured sensitivities (normalized counts per ppb; ncts/ppb), limits of detection (ppt), and linearity (R$^2$) for compounds used for calibration of both the GC-EI-TOF and GC-Vocus. RT-Vocus sensitivities (normalized counts per second per ppb; ncps/ppb) for a subset of the compounds are included for comparison.**

|  | Quant. Ion | | Sensitivity | | | LOD | | R$^2$ | |
|---|---|---|---|---|---|---|---|---|---|
|  |  |  | (ncps/ppb) | (ncts/ppb)[a] | | (ppt)[a,b] | |  |  |
|  | GC-EI-TOF | GC-Vocus | RT-Vocus | GC-EI-TOF | GC-Vocus | GC-EI-TOF | GC-Vocus | GC-EI-TOF | GC-Vocus |
| α-pinene | $C_7H_9+$ | $C_{10}H_{17}+$ |  | 23600 | 6920 | 0.2 | 0.4 | 0.98 | 1.00 |
| camphene | $C_7H_9+$ | $C_{10}H_{17}+$ |  | 20500 | 9690 | 1.0 | 1.2 | 0.99 | 1.00 |
| β-pinene | $C_7H_9+$ | $C_{10}H_{17}+$ |  | 10600 | 3850 | 1.7 | 2.6 | 0.98 | 1.00 |
| carene | $C_7H_9+$ | $C_{10}H_{17}+$ |  | 11500 | 3830 | 1.4 | 2.3 | 0.97 | 1.00 |
| limonene | $C_7H_9+$ | $C_{10}H_{17}+$ | 139 | 5300 | 2990 | 3.8 | 3.9 | 0.98 | 1.00 |
| benzene | $C_6H_6+$ | $C_6H_7+$ | 62 | 36800 | 3810 | 2.8 | 2.9 | 0.97 | 1.00 |
| toluene | $C_7H_7+$ | $C_7H_9+$ | 138 | 52700 | 9460 | 0.9 | 0.6 | 0.99 | 1.00 |
| ethyl-benzene | $C_7H_7+$ | $C_8H_{11}+$ |  | 43000 | 4680 | 1.6 | 1.5 | 0.98 | 1.00 |
| m&p-xylenes | $C_7H_7+$ | $C_8H_{11}+$ |  | 32100 | 12500 | 0.5 | 0.1 | 0.98 | 1.00 |
| o-xylene | $C_7H_7+$ | $C_8H_{11}+$ | 171 | 29300 | 9500 | 2.2 | 1.0 | 0.98 | 1.00 |

[a] Sensitivity and LOD for a sample volume of 180 cm$^3$, the volume used for GC-TOF ambient sampling during the ATHLETIC campaign.
[b] LOD calculated as 3 times the standard deviation of the baseline multiplied by the full width half maximum (FWHM) of the chromatographic peak divided by the sensitivity.

## 3 Results and Discussion

### 3.1 Benefits of Dual-Detector System and Instrument Performance

The novelty of this system is the ability to produce three complementary data sets: (1) RT-Vocus, (2) GC-Vocus, and (3) GC-EI-TOF during routine operation. As shown in Figure 1A, this setup allowed the Vocus to sample in real time from the weight room, the supply air, or the GC effluent. With this instrument configuration, the Vocus can make real-time measurements with fast time resolution (< 1 Hz) or automatically switch to GC detection for molecular speciation. When the Vocus was not sampling from the GC, the column effluent was instead sent to the EI-TOF for detection (shown in Figure 1A), which allowed continuous coverage of the GC identification measurements.

Another benefit of this system is that the two detectors use different ionization methods: proton transfer reaction (PTR) versus electron ionization (EI). By alternating between the detection methods, two sets of chromatograms were created with different



types of information about each molecule (which had the same GC retention time regardless of the detector). For GC-Vocus

chromatograms, the analytes are ideally detected as their protonated molecular ion [MH$^+$]. The extent to which this is true depends on instrument operating parameters such as the reduced electric field (*E/N*), where greater ratios tend to induce fragmentation due to increased collisions while also limiting the formation of cluster ions (Yuan et al., 2017). For GC-EI-TOF chromatograms, the analytes are detected by their ion fragments and identified with their EI fragmentation pattern. Benefits of EI detection, compared to PTR, is compound identification through fragmentation pattern matching (e.g. NIST/EPA/NIH mass

spectral library, Linstrom and Mallard, 2019, https://webbook.nist.gov/chemistry/) and the ability to detect compounds such as saturated hydrocarbons, which have a proton affinity too low for detection by PTR-MS (Yuan et al., 2017). However, since the Vocus generally detects an intact molecular ion, this can lead to a simpler analysis, where each compound is ideally detected as a single ion (giving the molecular formula of the protonated parent) rather than a series of fragments as with EI detection.

Calibration results (measured sensitivities, LODs, and R$^2$ values) from both the GC-EI-TOF and GC-Vocus are reported in

Table 1. A quantification ion was chosen for each method: for GC-Vocus this was the protonated molecular ion [MH$^+$], and for GC-EI-TOF the most abundant fragment ion present in the mass spectrum was used. These ions were chosen because they generally result in the highest sensitivity due to their abundance; however, in select circumstances (e.g. interference such as co-elution) another ion may be chosen. The comparison of GC-EI-TOF and GC-Vocus sensitivities and LODs is a comparison of detector response since GC operation was identical for calibration of each detector. EI-TOF detection was on average 4.3

times more sensitive than detection by the Vocus. This increase in sensitivity implies higher ion counts at the detector for the EI-MS versus PTR-MS; this may be attributed to several parameters including ionization efficiency (Cappellin et al., 2012; Sekimoto et al., 2017; Harland and Vallance, 1997).

Available RT-Vocus sensitivities are also reported in Table 1 for comparison with those measured by the GC-Vocus. For the limited overlap of calibration compounds presented in this work, the GC-Vocus was on average 50 times more sensitive than

the RT-Vocus. The gained sensitivity with the addition of the GC is due to differences in sample volume. When the Vocus is operating in the RT-Vocus mode, it samples ambient air at 100 cm$^3$ min$^{-1}$ with 1 s resolution, so each data point is representative of 1.7 cm$^3$ of ambient air. However, sensitivity is gained with the addition of the GC due to preconcentration of sample, where 180 cm$^3$ of ambient air is preconcentrated over the 6 min sample collection period before injection into the Vocus with typical peak widths here of ≤ 2 s (FWHM). The measured 50-fold increase in sensitivity agrees very well with the expected increase

of a factor of 53, which is calculated by dividing the 180 cm$^3$ sample preconcentrated by the GC and injected into the Vocus over a 2 s wide chromatographic peak (90 cm$^3$ s$^{-1}$) by the 1.7 cm$^3$ s$^{-1}$ analyzed by the RT-Vocus. The factor of 50 increase in sensitivity is due to the sampling schemes employed for this study. However, RT-Vocus LODs can be improved by averaging the signal to reduce noise. Figure S2 shows the Allan variance plot for the RT-Vocus C$_{10}$H$_{17}^+$ signal (protonated molecular ion of monoterpenes) during a relatively unperturbed, low concentration sampling period of room air. The Allan variance plot

shows a broad minimum around 250 s, indicating the maximum period of effective sample averaging; assuming Poisson



statistics for the data, this averaging window reduces RT-Vocus noise by a factor of ~ 16. This analysis shows that averaging the RT-Vocus data beyond 250 s would not decrease the noise further, as the RT-Vocus signal no longer follows Poisson statistics for these larger time periods, likely due to environmental factors such as changes in instrument temperature.

The GC LODs reported in Table 1 are a function of the standard deviation of the baseline surrounding the chromatographic
peak, chromatographic peak width, and instrument sensitivity. Across these compounds, the LODs for each instrument are very similar, each averaging ~1.6 ppt for a 180 cm$^3$ sample (GC-EI-TOF LOD$_{avg}$ = 1.6 +/- 1.1 ppt; GC-Vocus LOD$_{avg}$ = 1.7 +/- 1.2 ppt). As discussed in Section 2.8, the GC-Vocus measurements include a 100 cm$^3$ min$^{-1}$ make-up flow for instrument operation. For this study, ultra zero air was used for the make-up flow, as a gas with low purity will create elevated baselines and negatively impact the instrument LODs.

For each detection method, the differences in sensitivity and LODs between isomers (e.g. α-pinene versus limonene) are primarily a function of the extent to which the compound fragments. The greatest sensitivity would occur if the ionization of an analyte resulted in zero fragmentation, so that all analyte signal was associated with a single ion. Instead, if a compound fragments, the signal associated with that analyte is spread out across multiple ions, taking signal away from the ion used for quantification and decreasing the sensitivity (and increasing the LOD). In the α-pinene versus limonene example given above,
α-pinene fragments less than limonene, resulting in a greater sensitivity (and lower LOD) to the α-pinene monoterpene isomer.

**3.2 Molecular Identification of VOCs in an Indoor Environment**

The GC dataset was extensive and included detection of hydrocarbons, oxygenates and halocarbons in the volatility range of C$_5$–C$_{12}$ $n$-alkanes. Table 2 reports the EI characteristic ion and the ion(s) detected by the Vocus (typically a combination of the molecular ion [MH$^+$] along with water clusters and/or fragments) for a subset of the chromatographic peaks that were identified
through the GC analysis. These identified species include alcohols, aldehydes, ketones, ethers, nitrogen containing compounds, halocarbons, siloxanes, alkanes, alkenes, and aromatics. Specific results from the GC-TOF-MS system are reported in detail in Sections 3.3 for monoterpenes and in section 3.4 for dimethylsilanediol.

Table 2 presents three types of information about each molecule resolved by the GC: the retention time (which is a function of its vapor pressure, polarity, and functionality), and mass spectrometric response from both the EI-TOF and Vocus. One way
to identify compounds is to analyze standards, where each standard compound is injected into the GC to directly measure the analyte retention time and detector response. However, to do this type of analysis for each compound present in ambient air is time consuming and may not be feasible as some compounds are not available for purchase as analytical standards. Alternatively, retention time indices along with the mass spectrometric data can be used to confidently identify compounds without authentic standards. For example, a compound that eluted from the GC with retention time 369 s was detected in the
Vocus as both C$_7$H$_6$OH$^+$ and C$_7$H$_8$O$_2$H$^+$, which are formulas that could correspond to the protonated molecular ion of either benzaldehyde or methyl-benzenediols, respectively. However, the EI fragmentation pattern of this compound showed large



**Table 2. Compounds identified during the ATHLETIC campaign using the GC analysis with both EI-TOF-MS and Vocus PTR-TOF-MS detection.**

| Name | Retention Time (s) | EI-TOF-MS Characteristic Ion | Vocus PTR-TOF-MS Detected Ion | | | | | |
|---|---|---|---|---|---|---|---|---|
| | | | MH⁺ | M[H₂O]H⁺ | M[H₂O]₂H⁺ | Fragment | Fragment | Fragment |
| **Alcohols** | | | | | | | | |
| methanol | 153 | $CH_3O^+$ | $CH_4OH^+$ | $CH_4O[H_2O]H^+$ | $CH_4O[H_2O]_2H^+$ | | | |
| ethanol | 159 | $CH_3O^+$ | $C_2H_6OH^+$ | $C_2H_6O[H_2O]H^+$ | $C_2H_6O[H_2O]_2H^+$ | | | |
| 2-hexen-1-ol | 270 | $C_3H_5O^+$ | $C_6H_{12}OH^+$ | $C_6H_{12}O[H_2O]H^+$ | | $C_6H_{11}^+$ | | |
| nonen-1-ol | 424 | $C_5H_{10}^+$ | $C_9H_{18}OH^+$ | $C_9H_{18}O[H_2O]H^+$ | | $C_9H_{17}^+$ | $C_6H_{11}^+$ | $C_5H_9^+$ |
| decen-1-ol | 469 | $C_5H_{10}^+$ | $C_{10}H_{20}OH^+$ | $C_{10}H_{20}O[H_2O]H^+$ | | $C_6H_{11}^+$ | | |
| **Aldehydes** | | | | | | | | |
| acetaldehyde | 151 | $C_2H_4O^+$ | $C_2H_4OH^+$ | $C_2H_4O[H_2O]H^+$ | $C_2H_4O[H_2O]_2H^+$ | | | |
| 2-methyl propanal | 182 | $C_4H_8O^+$ | $C_4H_8OH^+$ | $C_4H_8O[H_2O]H^+$ | | | | |
| methacrolein | 185 | $C_4H_6O^+$ | $C_4H_6OH^+$ | | | | | |
| 2-butenal | 216 | $C_4H_6O^+$ | $C_4H_6OH^+$ | | | | | |
| furfural | 302 | $C_5H_4O_2^+$ | $C_5H_4O_2H^+$ | $C_5H_4O_2[H_2O]H^+$ | | | | |
| n-heptanal | 320 | $C_5H_{10}^+$ | $C_7H_{14}OH^+$ | $C_7H_{14}O[H_2O]H^+$ | | $C_7H_{13}^+$ | | |
| benzaldehyde | 369 | $C_6H_5^+$ | $C_7H_6OH^+$ | $C_7H_6O[H_2O]H^+$ | | | | |
| nonanal | 408 | $C_7H_{14}^+$ | $C_9H_{18}OH^+$ | $C_9H_{18}O[H_2O]H^+$ | | | | |
| decanal | 441 | $C_5H_{10}^+$ | $C_{10}H_{20}OH^+$ | $C_{10}H_{20}O [H_2O]H^+$ | | | | |
| **Ketones** | | | | | | | | |
| acetone | 166 | $C_3H_6O^+$ | $C_3H_6OH^+$ | $C_3H_6O[H_2O]H^+$ | | | | |
| methyl vinyl ketone | 192 | $C_4H_6O^+$ | $C_4H_6OH^+$ | $C_4H_6O[H_2O]H^+$ | | | | |
| methyl ethyl ketone | 194 | $C_4H_8O^+$ | $C_4H_8OH^+$ | $C_4H_8O[H_2O]H^+$ | | | | |
| 6-methyl-5-hepten-2-one | 368 | $C_8H_{12}^+$ | $C_8H_{14}OH^+$ | | | $C_8H_{13}^+$ | | |
| **Nitrogen Containing** | | | | | | | | |
| acetonitrile | 174 | $C_2H_3N^+$ | $C_2H_3NH^+$ | | | | | |
| *unknown* | *253* | | *$C_4H_5NH^+$* | *$C_4H_5N[H_2O]H^+$* | *$C_4H_5N[H_2O]_2H^+$* | | | |
| *unknown* | *268* | | *$C_4H_5NH^+$* | | | | | |
| **Halocarbons** | | | | | | | | |
| carbon tetrachloride | 204 | $CCl_3^+$ | ND | | | | | |
| tetrachloroethylene | 264 | $C_2Cl_3^+$ | ND | | | | | |
| parachlorobenzotrifluoride | 292 | $C_7H_4ClF_3^+$ | ND | | | | | |
| bromoform | 321 | $CHBr_2^+$ | ND | | | | | |





**340 Table 2 Continued. Compounds identified during the ATHLETIC campaign using the GC analysis with both EI-TOF-MS and Vocus PTR-TOF-MS detection.**

| Name | Retention Time (s) | EI-TOF-MS Characteristic Ion | Vocus PTR-TOF-MS Detected Ion | | | | | |
| --- | --- | --- | --- | --- | --- | --- | --- | --- |
| | | | $MH^+$ | $M[H_2O]H^+$ | $M[H_2O]_2H^+$ | Fragment | Fragment | Fragment |
| **Ethers** | | | | | | | | |
| furan | 166 | $C_4H_4O^+$ | $C_4H_4OH^+$ | | | | | |
| eucalyptol | 382 | $C_3H_7^+$ | $C_{10}H_{18}OH^+$ | | | | | |
| **Siloxanes** | | | | | | | | |
| dimethylsilanediol | 230 | $CH_5O_2Si^+$ | $C_2H_8O_2SiH^+$ | $C_2H_8O_2Si[H_2O]H^+$ | | | | |
| D3 siloxane | 252 | $C_5H_{15}O_3Si_3^+$ | $C_6H_{18}O_3Si_3H^+$ | $C_6H_{18}O_3Si_3[H_2O]H^+$ | | | | |
| D4 siloxane | 328 | $C_7H_{21}O_4Si_4^+$ | $C_8H_{24}O_4Si_4H^+$ | $C_8H_{24}O_4Si_4[H_2O]H^+$ | | | | |
| D5 siloxane | 411 | $C_9H_{27}O_5Si_5^+$ | ND | | | | | |
| **Alkanes** | | | | | | | | |
| n-hexane | 178 | $C_4H_9^+$ | ND | | | | | |
| methyl-cyclopentane | 193 | $C_4H_8^+$ | $C_6H_{12}H^+$ | | | $C_5H_{11}^+$ | $C_4H_9^+$ | |
| **Alkenes** | | | | | | | | |
| isoprene | 164 | $C_5H_8^+$ | $C_5H_8H^+$ | | | $C_5H_8^+$ | $C_5H_7^+$ | |
| α-pinene | 322 | $C_7H_9^+$ | $C_{10}H_{16}H^+$ | | | $C_6H_9^+$ | | |
| camphene | 334 | $C_7H_9^+$ | $C_{10}H_{16}H^+$ | | | $C_6H_9^+$ | | |
| β-pinene | 347 | $C_7H_9^+$ | $C_{10}H_{16}H^+$ | | | $C_6H_9^+$ | | |
| carene | 363 | $C_7H_9^+$ | $C_{10}H_{16}H^+$ | | | $C_6H_9^+$ | | |
| limonene | 374 | $C_7H_9^+$ | $C_{10}H_{16}H^+$ | | | $C_6H_9^+$ | | |
| γ-terpinene | 384 | $C_7H_9^+$ | $C_{10}H_{16}H^+$ | | | $C_6H_9^+$ | | |
| **Aromatics** | | | | | | | | |
| benzene | 211 | $C_6H_6^+$ | $C_6H_6H^+$ | | | $C_6H_6^+$ | | |
| toluene | 250 | $C_7H_7^+$ | $C_7H_8H^+$ | | | $C_7H_8^+$ | | |
| ethyl-benzene | 293 | $C_7H_7^+$ | $C_8H_{10}H^+$ | | | $C_6H_7^+$ | | |
| m&p-xylenes | 297 | $C_7H_7^+$ | $C_8H_{10}H^+$ | | | | | |
| o-xylene | 311 | $C_7H_7^+$ | $C_8H_{10}H^+$ | | | | | |
| styrene | 312 | $C_8H_8^+$ | $C_8H_8H^+$ | | | | | |
| n-propyl benzene | 340 | $C_7H_7^+$ | $C_9H_{12}H^+$ | | | | | |
| 1-ethyl-3&4-methyl benzenes | 344 | $C_8H_9^+$ | $C_9H_{12}H^+$ | | | | | |
| 1,3,5-trimethyl benzene | 347 | $C_8H_9^+$ | $C_9H_{12}H^+$ | | | | | |
| 1-ethyl-2-methyl benzene | 357 | $C_8H_9^+$ | $C_9H_{12}H^+$ | | | | | |
| 1,2,4-trimethyl benzene | 364 | $C_8H_9^+$ | $C_9H_{12}H^+$ | | | | | |
| 1,2,3-trimethyl benzene | 383 | $C_8H_9^+$ | $C_9H_{12}H^+$ | | | | | |



signal at $C_7H_6O^+$ and $C_6H_5^+$, which match the pattern expected for benzaldehyde ($C_7H_6O^+$, molecular ion; $C_6H_5^+$, due to loss of the aldehyde group). This information is enough to infer that the peak is benzaldehyde, and that the ions detected by the Vocus were the protonated molecular ion ($C_7H_6OH^+$) and its water cluster ($C_7H_6O[H_2O]H^+$). However, this assignment can be

made unambiguous by also comparing the retention time and retention index to other compounds present in the chromatogram (Van den Dool and Kratz, 1963; Linstrom and Mallard, 2019, https://webbook.nist.gov/chemistry/). The compound retention time of 369 s is between those of o-xylene (311 s) and limonene (374 s), which have retention indices of 890 and 1030, respectively (Linstrom and Mallard, 2019, https://webbook.nist.gov/chemistry/). Because benzaldehyde has a retention index of 960, we can predict it would elute between o-xylene and limonene, whereas methyl-benzenediols have retention indices of

~1200, meaning these would elute after limonene. With this method, we can use the combination of mass spectrometric data from the Vocus and the EI-TOF along with the chromatographic retention time to definitively identify compounds. Most species were positively identified using the workflow demonstrated for benzaldehyde; however, some compounds were only detected by a single detector. Unsaturated alkanes (e.g. *n*-hexane) and halocarbons (e.g. carbon tetrachloride, bromoform) were only detected by the EI-TOF, which was expected as these classes of compounds have proton affinities too low for detection

by PTR-MS methods (Yuan et al., 2017).

Aside from using the GC to aid in compound identification, the GC data can also be used to characterize the ion chemistry occurring in the Vocus.  From the information reported in Table 2, it can be seen that classes of compounds showed similar responses in the Vocus (e.g. forming single or double water clusters ($M[H_2O]H^+$ or $M[H_2O]_2H^+$) or undergoing fragmentation). Both results are non-ideal as they complicate the mass spectrum and make quantification and interpretation of the PTR-MS

signals more difficult. Alcohols, aldehydes, ketones, siloxanes, and an unknown nitrogen-containing compound all formed significant water clusters within the Vocus ion-molecule reactor, while alkenes and aromatics demonstrated fragmentation. Classes of compounds that were exclusively observed as their protonated molecular ion in the Vocus include ethers, large aromatics (e.g. $C_9$ aromatics) and some nitrogen containing compounds. The extent to which species form clusters or undergo fragmentation in the PTR-MS is a function of instrument operational parameters like the *E/N* ratio (operated here at 150 Td).

Although these artifacts complicate the RT-Vocus interpretation, with the addition of GC separation the molecular identification, and identification of fragment and cluster signals, is possible to do for complex ambient samples. Furthermore, by using GC separation to quantify the ratio of fragmentation or cluster formation to the protonated molecular ion, the RT-Vocus measurements can be constrained and more easily interpreted.

### 3.3 Monoterpenes in an Indoor Environment

### 3.3.1 Quantitative Assessment of Monoterpene Detection

A subset of the $C_{10}H_{17}^+$ time series, the monoterpene protonated molecular ion measured by RT-Vocus, is shown in Figure 2. During this period, GC-Vocus and GC-EI-TOF chromatograms were acquired and used to resolve individual monoterpene

**Figure 2. RT-Vocus detection of $C_{10}H_{17}^+$ with occasional speciation with GC-Vocus analysis. The RT-Vocus trace contains only room air measurements for simplicity. Pie charts show the GC-Vocus speciation of the RT-Vocus $C_{10}H_{17}^+$ signal into six resolved monoterpene species during each of the highlighted (grey) sampling periods. The pie chart fractions represent the contribution (in concentration, ppb) of each monoterpene isomer to the total monoterpene concentration measured by GC-Vocus. Some specific events are numbered and highlighted in light blue with the following distinctions: 1) exercise session, 2) in flow of outdoor air, 3) football game, and 4) post-game activities.**

isomers. The separation of the $C_{10}H_{17}^+$ RT-Vocus signal into six different monoterpenes by the GC-Vocus is shown as pie charts in Figure 2, where the pie chart fractions represent the contribution of each isomer (by concentration, ppb) to the summed





concentration of all monoterpene isomers resolved by the GC-Vocus. As mentioned above, when the GC effluent was not
being sent to the Vocus it was directed to the EI-TOF. Figure S3 shows the GC-EI-TOF time series of monoterpenes during

the same sampling period shown in Figure 2. During these sampling periods, limonene accounted for 47 – 80% of the measured
monoterpene composition due to activities occurring in and near the weight room. Details of the temporal behavior are
discussed in Section 3.3.2.

The RT-Vocus, GC-Vocus, and GC-EI-TOF monoterpene data sets are shown together in Figure 3. The RT-Vocus $C_{10}H_{17}^+$
signal along with the speciated monoterpenes (and their sum) resolved by GC-Vocus and GC-EI-TOF are overlaid in Figure

3A and 3B, respectively. The GC-EI-TOF monoterpene sum (Figure 3B) agrees within a factor of 1.2 on average to the RT-
Vocus signal across the entire measurement period. The GC-Vocus monoterpene sum also agrees within a factor of 1.2 with
the RT-Vocus signal on November 17; however, the agreement was not as good (only within a factor of 3) on November 16.
The better agreement on November 17 is due to the monoterpene composition being 70% limonene on average, which was the
isomer used to calibrate the RT-Vocus monoterpene signal.

Previous studies have found that comparing GC speciation with online PTR-MS measurements may result in discrepancies
(de Gouw et al., 2003; de Gouw and Warneke, 2007; Kari et al., 2018; Kim et al., 2009). For example, Kari et al. (2018)
demonstrated errors of 26% in PTR-MS ambient measurements when the terpene composition was not accounted for, and
instead a calibration factor determined from one isomer was used to represent the mixture. To avoid these errors, they urge
PTR-MS users to use complementary methods (e.g. GC) to identify the speciated terpene composition and calibrate the total

signal from PTR-MS using this speciation. The findings from Kari et al. (2018) are supported by the results shown here, where
the best agreement between the speciated (GC) and online (RT) measurements are when the monoterpene composition is
dominated by the isomer that was used to calibrate the RT-Vocus. When the mixture includes significant fractions of multiple
isomers, the bias in the RT-Vocus measurement increases. However, if RT-Vocus calibration factors for monoterpenes other
than limonene were measured the RT-Vocus signal could be weighted according to the continuous GC-EI-TOF speciation

measurements and corrected for the mixed monoterpene composition.

As discussed above, fragmentation and cluster formation can complicate the interpretation of real-time measurements, where
now there are multiple ions associated with one compound. An example of this behavior is shown in Figure S4, where Figure
S4A is a GC-Vocus chromatogram of the monoterpene protonated molecular ion, $C_{10}H_{17}^+$, and Figure S4B is the chromatogram
of $C_6H_9^+$, a common monoterpene fragment ion formed in proton-transfer reactions. Relative signals for the fragment ion

versus the protonated molecular ions for each monoterpene (labeled 1–6 in Figure S4A) are reported in Table 3. For all
monoterpenes, detection of the fragment ion $C_6H_9^+$ occurred in approximately a 1:1 ratio to the intact parent ion, $C_{10}H_{17}^+$.
Specifically, the ratios measured for the isomers ranged from 0.78 to 1.90 with an average of 1.15 +/- 0.40, these results are
comparable to monoterpene fragmentation ratios reported elsewhere (Steeghs et al., 2007). While the GC can be used to select





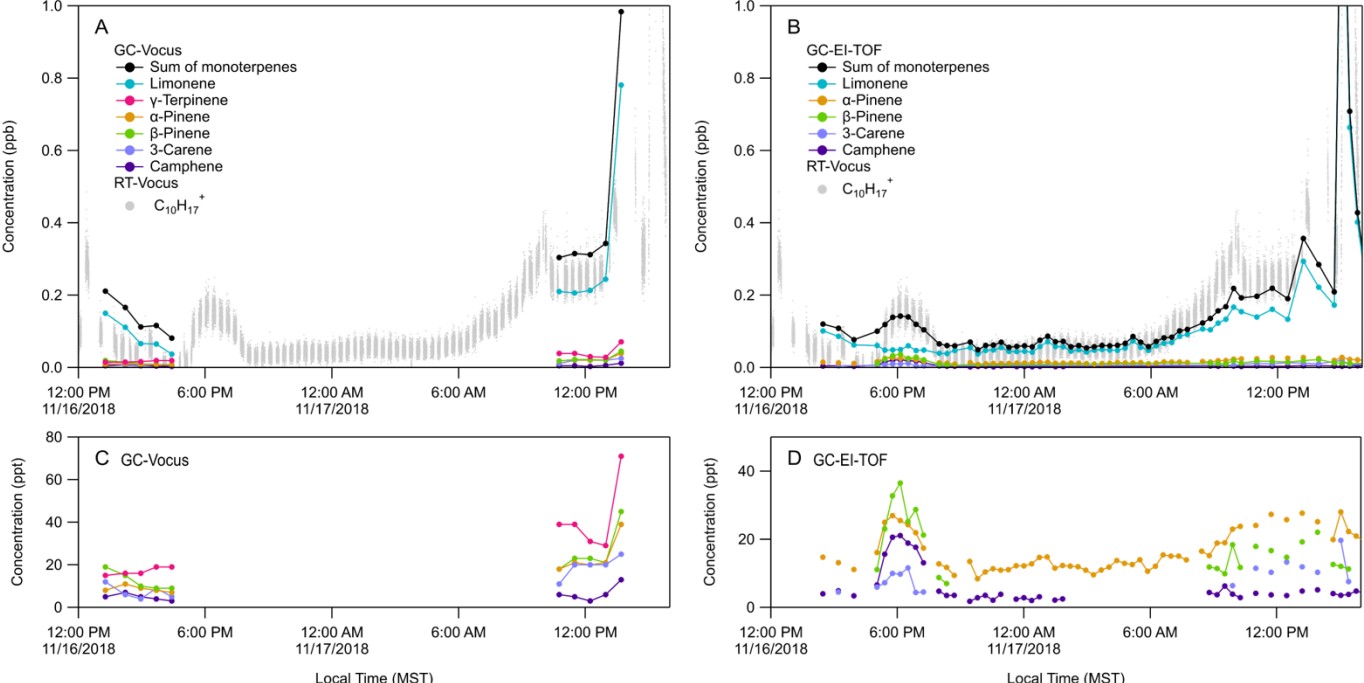

**Figure 3. Quantitative speciation of RT-Vocus $C_{10}H_{17}^+$ signal (grey trace) into resolved monoterpene isomers by (A) GC-Vocus (B) GC-EI-TOF. Panel C and Panel D show the same GC-Vocus and GC-EI-TOF timeseries as in Panel A and Panel B, respectively, but focused on isomers detected at lower concentrations. Absent data points in Panel D are for chromatograms where the chromatographic peak area was below the limit of detection. RT-Vocus trace in Panels A and B only shows RT-Vocus $C_{10}H_{17}^+$ detected in the room (not the supply air) for a direct comparison with the GC samples.**

PTR-MS parameters that optimize the formation of the protonated molecular ion, these other pathways are unavoidable across the range of compound classes observed, and routine GC measurements allow the user to account for them.

### 3.3.2 Monoterpene Temporal Behavior During ATHLETIC

During ATHLETIC, several monoterpene enhancement events took place while the system operated in multiple modes. On November 16, from 9 am to 12 pm local time, the RT-Vocus measured an increase in the $C_{10}H_{17}^+$ signal in the room air versus the supply air (Figure 2). This increase correlated with the presence of people in the weight room, who presumably acted as the source of the elevated levels of monoterpenes (likely from personal care products). The GC measurements between 1 pm to 4:30 pm on November 16 were taken while people were present in, and then left, the weight room. During this time the RT-Vocus $C_{10}H_{17}^+$ signal and the limonene measured by GC-Vocus decreased while the other monoterpene isomers stayed relatively constant (Figure 3A, 3C). Interestingly, at about 5 pm on November 16, the sum of monoterpenes measured by the



**Table 3. Relative signals of ions detected in the Vocus, fragmentation ion ($C_6H_9^+$) versus protonated molecular ion ($C_{10}H_{17}^+$), for monoterpene isomers.**

|  |  | Vocus Relative Signal |
| --- | --- | --- |
|  | Peak Number | $C_6H_9+/C_{10}H_{17}+$ |
| α-pinene | 1 | 0.97 ± 0.09 |
| Camphene | 2 | 1.90 ± 0.65 |
| β-pinene | 3 | 1.10 ± 0.41 |
| Carene | 4 | 1.13 ± 0.29 |
| Limonene | 5 | 1.00 ± 0.13 |
| γ-Terpinene | 6 | 0.78 ± 0.12 |

GC-EI-TOF showed an increase that was also observed in the RT-Vocus $C_{10}H_{17}^+$ signal (Figure 3B), and the GC speciation showed that the concentration of all the monoterpene isomers other than limonene increased (Figure 3D). The increase was first observed by the RT-Vocus in the supply air (Figure S5A), which, when combined with the GC-EI-TOF identification of non-limonene monoterpenes, allows us to attribute this event to an inflow of outdoor air not influenced by game-related activities.

An increase on November 17, between 9 am and 3 pm, occurred during a tailgating event and football game that took place adjacent to the Dal Ward Athletic Center. During this event, the room air and the supply air both showed increased $C_{10}H_{17}^+$ signals (Figure S5B) indicating a source in close proximity to the facility. Since the supply air for the weight room enters the building from an intake on the north side of the football stadium, VOCs emitted during the sporting event could be subsequently transported through the HVAC system and into the weight room. The large spike in $C_{10}H_{17}^+$ observed that day at 3 pm,

however, was predominantly measured inside the weight room rather than in the supply air (Figure S5C), suggesting an emission source in the room associated with people and possibly cleaning activities in the Dal Ward center after the game. Evaluating the November 17 enhancement events with the GC measurements, it is clear that the increase in the RT-Vocus $C_{10}H_{17}^+$ signal during the football game and the spike that occurred at 3 pm were due to an increase in limonene (Figure 3A, 3B), with no corresponding significant increases in the other monoterpene isomers (Figure 3C, 3D). The large monoterpene





enhancement observed during and after the football game, which was dominated by limonene, was likely due to VCPs from personal care products used by the athletes and spectators, and/or cleaning supplies.

### 3.3.3 Anthropogenic Signature of Monoterpenes

Indoor environments generally have relatively low oxidant and high VOC concentrations compared to the ambient atmosphere (Pagonis et al., 2019; Price et al., 2019). As a result, the fraction of monoterpenes that are oxidized indoors is small, and to a

large extent are transported outdoors. The monoterpene composition measured in this study provides anthropogenic source signatures that differ from those associated with typical outdoor biogenic sources (Guenther et al., 2012). These source signatures were compared using three ratios: limonene/α-pinene (lim/αp), limonene/β-pinene (lim/βp), and α-pinene/β-pinene (αp/βp). Generally, these ratios are a function of their emission profiles either from plant species outdoors, VCPs from human activity indoors, and a mixture of these sources in urban areas. Geron et al. (2000) reports monoterpene compositions for

forested regions in the United States, which give the following averages for these ratios (average +/- standard deviation): lim/αp = 0.32 +/- 0.21, lim/βp = 0.56 +/- 0.46 and αp/βp = 1.63 +/- 0.70. These values reflect the emission profiles from the types of plants in the forest and agree very well with those of Faiola et al. (2015), where the average ratios determined from emissions from coniferous plants (e.g. blue spruce, grand fir, bristlecone pine) were lim/αp = 0.29 +/- 0.16, lim/βp = 0.73 +/- 1.16, αp/βp = 2.18 +/- 2.28. These studies show that when the source of the monoterpenes is biogenic, α- and β-pinene

emissions dominate over limonene, and α-pinene emissions are about twice those of β-pinene. For comparison, the average ratios measured by GC-EI-TOF during ATHLETIC were lim/αp = 4.67 +/- 4.97 (maximum 56.4, minimum 1.62), lim/βp = 8.66 +/- 16.1 (maximum 131, minimum 1.36) and αp/βp = 1.73 +/- 0.61 (maximum 3.93, minimum 0.70). The large standard deviations of the limonene to α- and β-pinene ratios are due to the large limonene emission event after the football game on November 17. Unlike the biogenic ratios described above, the results from the indoor environment show that limonene

emissions always dominate over α- and β-pinene emissions, although the magnitude is highly variable and depends on proximity to sources (like the application of cleaning or personal care products). The αp/βp ratio measured during ATHLETIC is indicative of biogenic sources, consistent with our attribution of these isomers in the weight room to transport by outdoor air, where their source was likely biogenic. This shift in dominance from isomers like α- and β-pinene to limonene can have important implications on ambient air quality due to differences in reactivity (Atkinson and Arey, 2003) and SOA formation

potential (Lee et al., 2006).

### 3.4 Detection of Dimethylsilanediol

In the GC-EI-TOF chromatograms, a well-resolved peak was observed at retention time 230 s (Figure S6A) with a mass spectrum base peak at $m/z$ 77 (Figure S7A). Using TERN analysis software, the unit mass resolution (UMR) electron ionization fragmentation pattern for this chromatographic peak (Figure S7) was compared against the NIST mass spectral database

(Linstrom and Mallard, 2019, https://webbook.nist.gov/chemistry/) resulting (90.8 % probability) in an identification of that





compound as dimethylsilanediol (DMSD). Analyzing the same chromatographic peak with the high-resolution EI-TOF data, the mass spectral peak at UMR $m/z$ 77 was assigned as molecular formula $CH_5O_2Si^+$ (Figure S8A). This characteristic ion is formed from DMSD through the loss of a methyl ($CH_3$) group. The protonated molecular ion of DMSD ($C_2H_8O_2SiH^+$) and its protonated water cluster ($C_2H_8O_2Si[H_2O]H^+$) were identified at the same retention time in the GC-Vocus chromatogram

(Figure S6B). The tailing observed only in the GC-Vocus chromatographic peak was likely from inefficient transport through the transfer line due to non-uniform heating. The high-resolution mass spectral fits for these peaks from the GC-Vocus are shown in Figures S8B and S8C, respectively. From the high-resolution identification of the molecular ion (GC-Vocus) and the fragmentation pattern with characteristic ions (GC-EI-TOF), this compound was conclusively identified as DMSD.

It should be recognized that siloxanes have historically been a class of compounds that are difficult to measure analytically

due to artifacts created throughout sampling systems (Rücker and Kümmerer, 2015). For example, siloxanes have been associated with artifacts for GC analyses that utilize septa in the sample path (de Zeeuw, 2005; Wang, 2006). For the instrument used in this study, no septa are present, and with our *in situ* GC sampling we avoid many artifacts that can be introduced due to sample collection and storage for off-line analysis. The sample trap used for pre-concentration is also of concern for artifact generation. To determine if DMSD is being produced as an artifact from the adsorbent trap, we conducted system blanks using

dry (RH < 2%) and humidified (RH ~ 50%) UHP $N_2$ to cover humidity conditions relevant for this study, and saw no evidence of DMSD. From these results, we conclude that the DMSD observed during this study was not an artifact of the instrumentation but was instead present in the air sampled from the weight room.

Figure 4 shows the time series of the integrated chromatographic peak areas detected from the GC-EI-TOF. The time series shows dynamic behavior where DMSD builds up in the room starting at midnight on November 17 until a sharp decay after

the football game, as seen with the RT-Vocus $C_{10}H_{17}^+$ signal (Figure 2) and the GC-EI-TOF limonene signal (Figure S3A). This elevated concentration of DMSD on November 17 is also observed in the more limited GC-Vocus data set where only 10 chromatograms were obtained. As shown in Figure S9, DMSD was detected as both the protonated molecular ion ($C_2H_8O_2SiH^+$) and as the protonated water cluster of the molecular ion ($C_2H_8O_2Si[H_2O]H^+$). After the decay on November 17, the DMSD stabilizes to a background concentration until another increase in concentration starting in early afternoon on

November 18. The temporal behavior follows that of the RH in the room (Figure 4), where DMSD begins to increase following an increase in RH and then declines sharply as the RH begins to decrease. Figure 4 shows that during the first DMSD enhancement, the fraction of outside air in the supply air was tapering down from 40% to 15%; however, during the enhancement on November 18, outside air was increasing from near 0 to about 20%. The lack of correlation between DMSD and the fraction of outdoor air in the supply air indicates that this compound was not being transported from outdoors, but

instead had a source inside the room or building. The sharp decrease in DMSD, which correlated with the observed decay for limonene on November 17, appears to be from reduced production of DMSD in the room along with fast loss from ventilation. The observed behavior of DMSD does not follow that of decamethylcyclopentasiloxane ($D_5$), a siloxane commonly used in

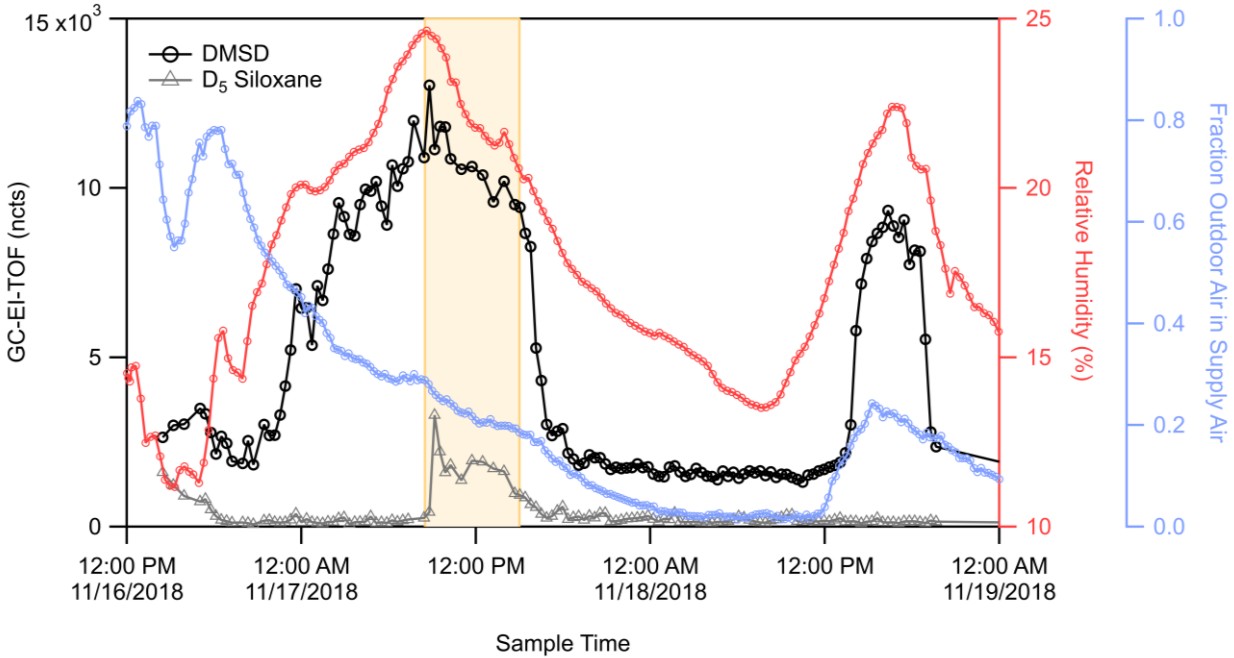

**Figure 4. Time series of DMSD (black trace) and D$_5$ siloxane (grey trace) detected by GC-EI-TOF (normalized counts, ncts), weight**
**room relative humidity (pink trace), and the fraction of outside air in the room supply air (blue trace). The area highlighted in yellow**
**represents the time during which pre-game activities and a football game were occurring adjacent to the athletic center.**

personal care products like deodorants (Coggon et al., 2018; Tang et al., 2015). The time trace for D$_5$ is also shown in Figure

4. There is an enhancement in the D$_5$ signal during the football game and associated activities on November 17 (due to people

inside and adjacent to the weight room), but the D$_5$ time series does not track the initial buildup of DMSD on November 17

and shows no enhancement during the DMSD pulse on November 18. Therefore, it is apparent there are different sources for

these two organosiloxane compounds.

DMSD has been shown to be an environmental degradation product of both cyclic (cVMS) and linear (PDMS) siloxanes

(Rücker and Kümmerer, 2015; Tuazon et al., 2000). Both classes of organosiloxanes degrade to DMSD through gas-phase

oxidation by hydroxyl (OH) radicals (Tuazon et al., 2000) and through condensed-phase hydrolysis reactions (Xu et al., 1998;

Lehmann et al., 1994a,b; Lehmann et al., 1995; Carpenter et al., 1995). While there is uncertainty surrounding the importance

of OH radical chemistry indoors, previous works have estimated that typical indoor OH radical concentrations are on the order

of $10^5$ molecules cm$^{-3}$ due to low lighting conditions that reduce conventional photolysis reactions that produce OH radicals

(Weschler and Carslaw, 2018; Abbatt and Wang, 2020; Pagonis et al., 2019; Gligorovski and Weschler, 2013; Young et al.,

2019). In the weight room during the DMSD enhancement events there was no natural light (i.e. no windows in the room) and







**Figure 5. Proposed mechanism for the formation of DMSD from the hydrolysis of PDMS.**

little artificial light as the space was often unoccupied. And, as discussed above and shown in Figure 4, DMSD enhancement
events do not track the observed behavior of $D_5$ siloxane. For these reasons, we conclude that the DMSD was not formed
through gas-phase oxidation of cVMSs (e.g. $D_5$ siloxane) by OH radicals, and instead hypothesize that the production is through
condensed-phase reactions followed by volatilization.

The mechanism by which PDMS (the less volatile class of organosiloxanes) decompose to form DMSD has primarily been
studied with regards to degradation in soils. There, PDMS undergoes moisture dependent hydrolysis to form DMSD, a process
shown to be catalyzed by minerals present in clay (Xu et al., 1998). Figure 5 provides a proposed mechanism for the formation
of DMSD from PDMS, though note that the PDMS co-product of DMSD can also undergo hydrolysis to produce more DMSD.
Once formed, the DMSD then either volatilizes or is retained in the condensed phase to undergo subsequent reactions.
Depending on the moisture conditions and mineral composition, hydrolysis occurs on timescales of minutes to days and
efficiently in the presence of calcified minerals like Ca-kaolinite, Ca-beidellite, and Ca-nontronite (Xu et al., 1998). The
hydrolysis of PDMS to DMSD has been shown to only occur in the presence of moderate amounts of water (Rücker and
Kümmerer, 2015; Lehmann et al., 1998). Some water is required for the hydrolysis reaction, but when moisture levels are too
high the reaction shuts off, presumably due to saturation of catalytic mineral sites by adsorbed water.

While the exact source of DMSD observed during this study cannot be determined from our measurements, we can suggest at
least two possible sources for the DMSD production that would account for our observations. As shown in Figure 6, the DMSD
signal has a sigmoidal relationship with RH, where the greatest rate of change in DMSD production is around 20% RH. The
steepness of the curve suggests the DMSD is being produced from a discrete process, likely due to some material that combines
siloxanes with a mineral catalyst taking up water around 20% RH providing the needed water for the hydrolysis reaction.





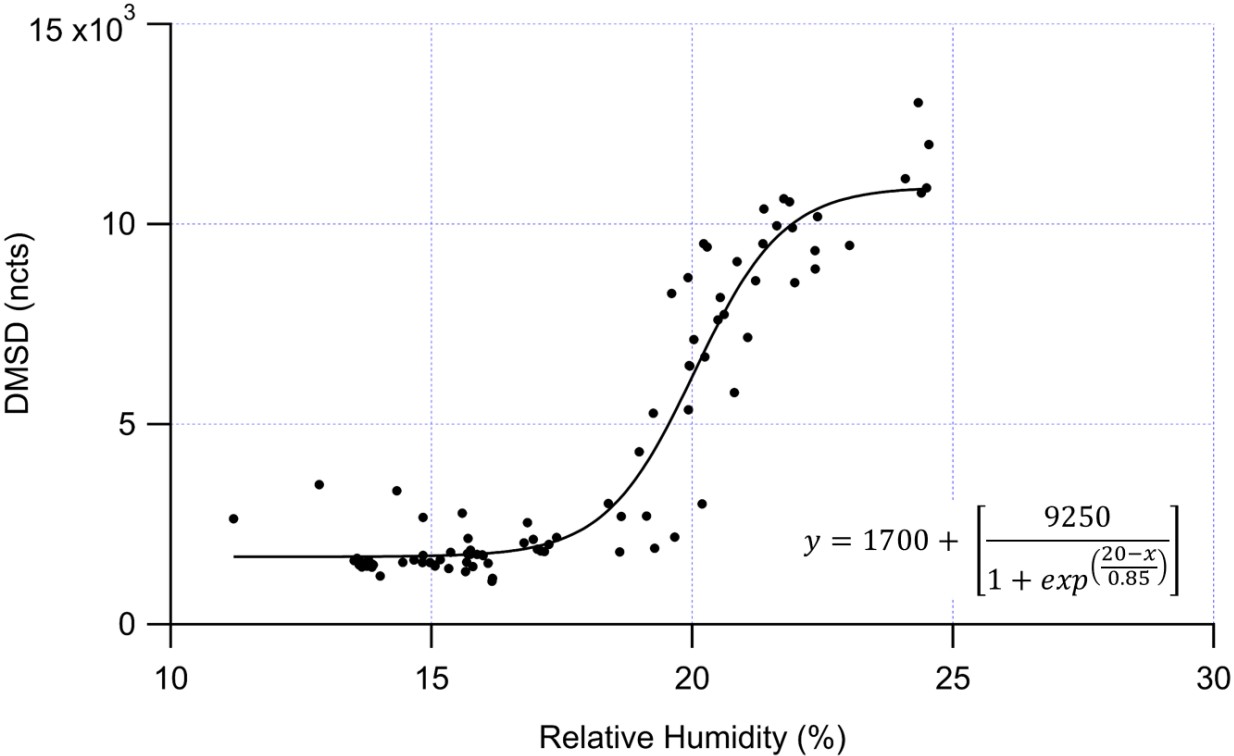

**Figure 6. DMSD (normalized counts, ncts) measured by the GC-EI-TOF versus relative humidity (%). The data was fit with a sigmoidal curve that shows the greatest rate of change in DMSD production at ~ 20% RH.**

Indoors, sources of PDMS are numerous: paints, coatings, sealants, textiles, and electronics (Andriot et al., 2007). Particularly relevant is their use in paints, where PDMS are used as the binder (30–100 % w/w of the mixture) and as additives (0.1–5 % w/w of the mixture), depending on the product and manufacturer (Andriot et al., 2007). Another significant component of paints are inorganic minerals, used as pigments and extenders, constituting 20–50% of the paint mixture (Karakaş et al., 2010). While $TiO_2$ is commonly added for its optical properties, minerals like calcite, Ca-kaolin, talc, and dolomite are used as fillers (Karakaş et al., 2011). The abundance of painted surfaces indoors and the correlation between room humidity and enhancement of DMSD (Figure 4) provide a possible explanation for the observation of DMSD during this study: mineral-catalyzed hydrolysis of PDMS on painted surfaces analogous to what occurs in soils. While the presence of PDMS in paints and coatings makes this chemistry plausible, the partitioning of lower-volatility cVMS could also act as a source of condensed-phase siloxanes in indoor environments. Using the $D_5$ siloxane vapor pressure measured by Lei et al. (2010), the saturation vapor concentration (C*) is estimated to be ~$10^6$ μg $m^{-3}$. According to Pagonis et al. (2019) and Algrim et al. (2020), for compounds


with $C^* < 10^8$ μg m$^{-3}$ deposition to surfaces can compete with ventilation as a removal process. Thus, some of the cVMS emitted in the weight room may partition to surfaces where they could react to produce DMSD as discussed above for PDMS.

The observation of DMSD as an atmospheric reaction product of cVMS, as a soil degradation product of PDMS, and now in an indoor environment where its formation is proposed to occur by surface hydrolysis of siloxanes, demonstrates the need to quantify its environmental impact. A review by Rücker and Kümmerer (2015) refers to DMSD as the most important intermediate of organosiloxanes degradation, and yet very little is known about its physical properties, even boiling point or vapor pressure. This lack of knowledge has been due in part to an absence of analytical methods that can measure these

compounds. Here, we have demonstrated an *in situ*, field-deployable technique capable of measuring DMSD that can be used in a wide variety of environments to understand the sources and fates of this compound. These measurements are of particular interest considering that the eventual fate of DMSD in the atmosphere is to form $CO_2$ and $SiO_2$ (sand), where $SiO_2$ could contribute to new particle formation and growth in the atmosphere (Bzdek et al., 2014).

## 4 Summary and Conclusions

In this study we present a field-deployable *in situ* GC with thermal desorption preconcentration and automatic switching between two time-of-flight mass spectrometric detectors, Vocus PTR-TOF-MS and EI-TOF-MS. We have demonstrated how this novel technique generates three complementary data sets: RT-Vocus for fast, non-speciated, detection of VOCs, and molecular identification with both GC-Vocus and GC-EI-TOF. The latter provide molecular ion and electron ionization fragmentation pattern information, respectively, for each compound resolved by the GC. Unambiguous molecular

identification is obtained with the combination of these three techniques and the molecular speciation from the GC methods can be used to aid interpretation of complex real-time PTR-MS measurements, where fragmentation and the formation of clusters can complicate analysis. By including the GC with thermal desorption preconcentration, Vocus sensitivities were increased by a factor of 50 on average over the real-time measurements that were acquired with 1 Hz time resolution, and both GC-TOF methods demonstrated LODs of 1.6 ppt on average across a range of monoterpenes and aromatics.

To demonstrate this technique, the prototype GC-TOF-MS system was deployed during the 2018 ATHLETIC campaign at the University of Colorado Dal Ward Athletic Center to characterize VOC profiles with detailed speciation and high time resolution. The results presented report the identification of a wide range of VOCs, including hydrocarbons, oxygenates, and nitrogen-containing compounds and their responses in each TOF-MS detector. Two sets of notable results from the GC-TOF-MS system are described in detail, including the quantification of speciated anthropogenic monoterpenes, where the

composition is dominated (47 – 80%) by limonene due to the use of personal care products and cleaning supplies in the indoor environment. Furthermore, the detection of DMSD, hypothesized to be due to the heterogenous hydrolysis of siloxanes on painted surfaces, demonstrates this technique's ability to detect new processes due to its ability to be field deployed with *in*

*situ* sampling, high time resolution measurements, and high-resolution mass spectrometric detection which provides molecular formulas to aid interpretation.

Further development including flow path optimization, expanding the volatility range of resolved VOCs, and increasing system sensitivities by preconcentrating larger sample volumes using multi-stage sample trapping is underway. Increased instrument sensitivity and decreased limits of detection scale with larger sample volumes, and while the simplified TDPC used here had limitations to sampling rate and volume, we have recently developed a two-stage TDPC that improves upon these parameters and can collect large volumes (1 L typical) in 10 min or less. Although the instrument presented here was a prototype system, 605 the results reported demonstrate that this is a valuable analytical tool that should be deployed in future field campaigns and laboratory experiments to characterize VOC emissions and their reaction products in new and changing environments.

**Author Contribution**

MSC and BML built the ARI GC designed by BML and JTJ. The ATHLETIC campaign was designed by PJZ, JLJ, JdG. Instrument operation during ATHLETIC: GC-TOF-MS by MSC, Vocus PTR-TOF-MS by DP, Piccaro by AVH and DAD. 610 Data analysis: GC-TOF-MS by MSC, RT-Vocus by ZF, Piccaro by AVH and DAD, temperature and air handling by WLB. MSC prepared the manuscript with contributions from all coauthors.

**Competing Interests**

The authors declare that they have no conflict of interest.

**Acknowledgements**

The authors acknowledge Jason DePaepe, Shawn Herrera, MT Eisner, Jennifer Green, and Jeremy Johnson at the University of Colorado - Athletics and Facilities for hosting and supporting this study. We thank the Alfred P. Sloan Foundation (Grants No. G-2016-7173 and G-2019-12444) for funding the ATHLETIC study and for contributing to the support of subsequent activities related to this publication. WLB acknowledges support from a CIRES Graduate Fellowship.

**Supporting Information**

Monoterpene calibration curves for GC-EI-TOF and GC-Vocus; monoterpene chromatogram by GC-Vocus; Allan variance plot of RT-Vocus data; monoterpene timeseries by GC-EI-TOF; RT-Vocus detection of $C_{10}H_{17}^+$ in the room versus supply air during elevated events; chromatogram, electron ionization mass spectrum, high resolution fitting, and time series of dimethylsilanediol.



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
