# Peer review of "An *in situ* gas chromatograph with automatic detector switching between Vocus PTR-TOF-MS and EI-TOF-MS: Isomer resolved measurements of indoor air"

_Atmospheric Measurement Techniques, 2020_

## Referee Comment (RC1) · Anonymous Referee #1 · 14 Aug 2020

This paper presents a new instrument taking advances of both chromatographic and direct MS methods and two different ionization systems having therefore high potential for producing new kind data future air chemistry studies. Manuscript is clearly suitable for AMT. The method is well-described. Detection limits are low enough not just for indoor/urban air, but also for measurements of ambient air at more remote sites. Here results from indoor air measurements were presented showing the great potential of the instrument. I recommend publishing with minor changes.

Specific comments:

[Figure]

Line 109: The range of compounds is not this large if you use the trap at 20C. For very light VOCs, like ethane and ethane, breakthrough volume even at -30C is quite low. Please, correct this.

Line 199-200: Did you flush whole 5 ml/min into the column or did you have some split? What was your desorption efficiency?

Line 245: You calibrated your system with 2 ppb standard and sampling of time of 1 to 6 min.. Lowest calibration corresponds the ambient air concentration of 333 pptv, which is clearly higher than your detection limit. Did you test the linearity of your calibration curve with lower concentrations? Sometimes with TD systems curve is not linear with lower concentration for all compounds (due to the incomplete desorption or other losses in the system). At least camphene results in Fig. S1 give some indication on this.

Section 2.7: You have quite long inlet line (3.4m) for this low flow (30 ml/min). Maybe for future prototypes you will increase the flow to enable the quantitative measurement of more sticky compounds as well.

Section 2.8: Could you add a chromatogram (calibration and indoor air) maybe as a supplement? It is very nice if with this short chromatogram, you are able to separate so many different compounds.

Section 2.8.: Did you detect any blank/background for any of the measured compounds? Degradation of Tenax TA results often to some blank (e.g benzene).

Table 1: Could you also give precision and uncertainty of these systems in this table or in some other part of the paper?

Table 2: Even though you are able to detect some compounds (e.g. methanol, acetaldehyde etc.) I doubt their measurements are not quantitative. I would expect high breakthrough of them from the cold trap. Also some other molecules may have high losses in TD. If you have some results that show they are quantitative, please present

it. If not, maybe you could mention more clearly that GC can also be used just for identification of these compounds and maybe RT-Vocus can be used for quantification of some of them?

Line 342: Benzaldehyde has often quite high background in TD-GC runs (possibly due to degradation of Tenax TA). Even though it does not matter here, it could be more appropriate to use some other compounds as an example.

Section 3.4: Did you detect this compound with RT-Vocus? If so, please, show the results. This would prove that this is not coming from the TD system.

Is DMSD known to have some health effects?

---

## Referee Comment (RC2) · Anonymous Referee #2 · 25 Oct 2020

Claflin et al. demonstrate a novel dual-channel TDPC-GC-EI/Vocus($H_3O^+$)TOFMS instrument. The combination of chromatography with preconcentration, soft and hard ionization methods, highly time and mass resolved time-of-flight detector makes it probably the most universal and comprehensive state-of-the-art instrument currently available for time-resolved isomer-speciated VOC measurements. These measurements are particularly needed in the field of air quality and indoor and outdoor atmospheric chemistry. This manuscript shows a significant improvement in quantified chemical completeness, time resolution, and molecular speciation thanks to enormous

synergy from coupling complementary state-of-the-art analytical methods. While I can see some potential for further improvement, overall, the manuscript is well written and, in my opinion, will be a valuable contribution to the fields of gas chromatography and mass spectrometry for indoor and other atmospheric applications. I would have relatively minor comments and suggestions which hopefully can be easily addressed in the revision.

**Specific Comments and Suggestions**

1) The title reads nicely exhaustively informative but the presence of "indoor air" in the title might be misleading. I interpret the novel instrument/method as more generally applicable than just for the indoor air but perhaps the title might mislead the AMT audience that the method/instrument is dedicated only to indoor air measurements rather than that the indoor air was just the indoor gym field example. The extremely impressive detection limit thanks to the Vocus sensitivity and preconcentration makes this method particularly powerful for discoveries also in the outdoor atmosphere and many other contexts.

2) I think the novel instrumentation presented in this paper is absolutely outstanding, but I do have a feeling that the capabilities are *much* greater than described in the manuscript. The table 2 nicely shows different classes but with the sub-ppt detection limit indoors one would expect thousands of ions. Are the compounds in Table 2 just select, example compounds from the weight room or was it meant to represent the complete chemical composition?

3) The paper focuses predominantly on monoterpenes, select aromatics and silicon-containing VOCs (cVMS, DMSD). This is great but I would recommend expanding beyond the weight room, on the detectable compounds, ideally across a range of c*, and chemical classes. It would also be nice to add to the discussion which compounds are not detectable or are particularly challenging.

4) I really appreciate the switching capability between the RT-Vocus, GC-Vocus and

[Figure]

GC-EITOF. The "automatic detector switching" is emphasized already in the title. However, I could not find information how fast the switching is and how the data between switching is treated/trimmed. It would be great to include this information.

5) One big issue, not specific to this paper, but applicable to analytical chemistry methods in general are potential chemical conversions in the instruments or sampling system due to contact with materials (e.g. metal surface) or thermal (e.g. high temperature ramp or desorption). The authors are in an excellent position to shed some light on this question because RT-Vocus and GC-Vocus data can be directly compared for compounds which would be expected thermally unstable. I think expanding on this general issue could be interesting for the AMT community.

6) 100 ppq LOD for mp-xylenes is certainly extremely impressive! While 1 ppt for o-xylene is still impressive, I wonder what exactly is causing a large difference in LOD between those isomers.

7) Two units are used for the normalized signals (ncps, and ncts). The text cautions the reader about the differences which helps. The normalization process is well described in Sect. 2.9. It seems that the ncps normalization was done by the second water cluster which makes me wonder if the signal was relatively constant in the Vocus at the given E/N ratio and unaffected by sample humidity. Is it assumed that this ion would reflect changes in H3O+ more than the changes in ambient H2O? Because changes in the E/N ratio would largely affect ncps values normalized to humidity-independent water cluster I would suggest adding a subscript with E/N ratio used (e.g. $Sn_{150Td}$). This should allow for comparisons in future campaigns and prevent confusion of normalized sensitivities derived at different E/N ratios. I would also suggest showing in addition (maybe in parenthesis) the absolute sensitivity (cps/ppb).

8) Further to the comment above, I have been missing some details on the Vocus operation and data processing. The reader is referred to the paper by Finewax et al. (2020) but this paper does not seem to be published yet so I could not refer to it. It

is great to see the parameters for the IMR, but it is unclear if the TPS voltages have been optimized with the Thuner or manually. I am also specifically wondering why 1.5 mbar of IMR pressure was used? It is not an issue but usually >=2 mbar is used. The E/N ratio of 150 Td is already somewhat high so the higher pressure could lower it and further boost the sensitivity but if there was a specific reason perhaps it could be interesting to include.

9) It is nice to see the good performance of the GC-(PTR)Vocus channel. For instance, the speciating power of monoterpenes looks simply excellent. In terms of the other isomeric mixtures, would there be any benefit from using GC-(NH4+)Vocus ionization or has it not been tried yet in this configuration? Perhaps it could be inspiring to add some prognosis on this to the future work.

10) By looking at the detected compounds in the indoor campaign (Table 2) I am missing more highly oxygenated compounds such as acids, hydroxy acids. Would it be useful to try the instrument with an in-situ derivatization (e.g. Isaacman et al., 2014)? Other compound families I am wondering about detection/speciation by this GC are sulfur-containing, amides, amines, heterocycles, metalorganics. By comparing the data from the GC-Vocus and RT-Vocus, it should be possible to delineate the groups of compounds which may not have made it through the column.

11) It is great to see GC and Vocus synergistically complementary. I understand that Finewax et al. (2020) is going to report expanded Vocus dataset from the gym, but I wonder why D5 is shown as detected by GC-EITOF but not GC-Vocus (Table 2). It is surprising because Vocus is definitely very sensitive to D5. Could it be that the sample did not reach Vocus for some reason? What was the detectability of D6 and D7?

12) L365 The use of "artifacts" term in this context reads extremely misleading here. The fragments or clusters are typically not artifacts in PTRMS. In many cases they can be used to quantify compounds (e.g. the cyclohexadiene fragment of monoterpenes m/z 81.0699 or methanol cluster m/z 51.0446). I suggest replacing with "interferences"

[Figure]

or "complications" to avoid confusion with artifacts from sampling tube materials, etc.

13) The DMSD story is very well done. Clearly this discovery would have been much more difficult without the complementary power of this instrument. However, I am completely unconvinced by the indoor OH radical hypothesis. It simply does not make sense to me in terms of Fig 4 showing increase in concentration over night and being correlated with RH. This does seem perfectly aligned with a possibility of microbial biodegradation of siloxanes in PCPs in sweat. It would be consistent with numerous sources reporting it as a biodegradation product (Accettola et al., 2008; Xu, 1999). While this explanation seems most likely to me for this indoor air case, it does not necessarily mean that DMSD is not formed via OH oxidation outdoors which would be another example of an analogy between the atmospheric and microbial oxidation/degradation.

14) The quantified speciation of monoterpenes by GC-Vocus is extraordinarily skillful. These instruments are perfectly suited to contribute to a progress in source apportionments between anthropogenic, plant, fruit, and microbial sources of this important group of compounds. I strongly suspect but it would be great to know if the instrument is also capable of speciating sesquiterpenes.

**Technical**

15) In several places a number and a unit are not separated by a space.

**References:**

Accettola, F., Guebitz, G.M. and Schoeftner, R., 2008. Siloxane removal from biogas by biofiltration: biodegradation studies. Clean Technologies and Environmental Policy, 10(2), pp.211-218.

Isaacman, G., Kreisberg, N.M., Yee, L.D., Worton, D.R., Chan, A.W.H., Moss, J.A., Hering, S.V. and Goldstein, A.H., 2014. Online derivatization for hourly measurements of gas-and particle-phase semi-volatile oxygenated organic compounds by thermal

desorption aerosol gas chromatography (SV-TAG). Atmospheric Measurement Techniques, 7(12).

Xu, S., 1999. Fate of cyclic methylsiloxanes in soils. 1. The degradation pathway. Environmental science technology, 33(4), pp.603-608.

---

## Author Comment (AC1) · 19 Nov 2020

This paper presents a new instrument taking advances of both chromatographic and direct MS methods and two different ionization systems having therefore high potential for producing new kind data future air chemistry studies. Manuscript is clearly suitable for AMT. The method is well-described. Detection limits are low enough not just for indoor/urban air, but also for measurements of ambient air at more remote sites. Here results from indoor air measurements were presented showing the great potential of the instrument. I recommend publishing with minor changes.

Specific comments:

Line 109: The range of compounds is not this large if you use the trap at 20C. For very light VOCs, like ethane and ethane, breakthrough volume even at -30C is quite low. Please, correct this.

Authors' response: The reviewer is correct that the range quoted here is mis-leading as it is the maximum range of the trap itself and not for the entire TDPC-GC system. We have updated the text to clarify, the text now reads:

"The combination of adsorbents in the TO-15/TO-17 trap allows for the analysis of a wide range of VOCs (including oxygenates) in the $C_2 - C_{32}$ n-alkane volatility range. However, for the system deployed for this work the instrument was operated in a way that was optimized for VOCs in the $C_5 - C_{12}$ volatility range. Details of operational parameters (e.g. temperatures, flows) are described in Section 2.8."

Line 199-200: Did you flush whole 5 ml/min into the column or did you have some split? What was your desorption efficiency?

Authors' response: We did not use a split flow for our injection, so the entire 5 sccm was sent to the column. To gauge our desorption efficiency, we would run a sample and then an instrument blank and measure the residual sample. The result of the instrument blank was < 1% of the signal measured in the sample. We have added the following text to Section 2.8 to address this:

"To gauge our desorption efficiency, we would run a sample and then an instrument blank, with no sample flow through the trap during the collection period, to measure the residual sample remaining in the trap. The result of the instrument blank was < 1 % of the signal measured in the sample indicating highly efficient transfer of sample, and this was deemed acceptable."

Line 245: You calibrated your system with 2 ppb standard and sampling of time of 1 to 6 min.. Lowest calibration corresponds the ambient air concentration of 333 pptv, which is clearly higher than your detection limit. Did you test the linearity of your calibration curve with lower concentrations? Sometimes with TD systems curve is not linear with lower concentration for all

compounds (due to the incomplete desorption or other losses in the system). At least camphene results in Fig. S1 give some indication on this.

Authors' response: The reviewer presents an excellent point, and for this study we did not calibrate to lower concentrations. The authors acknowledge the reviewers point about incomplete desorption and other losses impacting the lower end of the calibration curve. Since this study, we have transitioned to multi-stage trapping which has many benefits, making the single-stage TD configuration described in this paper obsolete. With our multi-stage trapping configuration, we have increased the dynamic range of our calibrations to over 2 orders of magnitude (down to 0.2 ppb). We are continuing to expand this range to lower concentrations. But until we can calibrate to lower concentrations, we acknowledge larger uncertainties at the lower mixing ratios.

Section 2.7: You have quite long inlet line (3.4m) for this low flow (30 ml/min). Maybe for future prototypes you will increase the flow to enable the quantitative measurement of more sticky compounds as well.

Authors' response: We agree with the reviewer. For campaigns since this work we have implemented a "fast inlet" which uses ¼" OD PFA tubing with an external pump, with a short 1/8" PFA line via tee to the GC inlet; this new inlet typically has a total residence time of <1s from the inlet tip to GC port. The GC system has also improved, and we now sample at much faster flows (100-150 sccm typical).

Section 2.8: Could you add a chromatogram (calibration and indoor air) maybe as a supplement? It is very nice if with this short chromatogram, you are able to separate so many different compounds.

Authors' response: We have added an example calibration and indoor air chromatogram from the GC-Vocus to demonstrate the separation of monoterpenes and $C_7$ and $C_8$ aromatics during this project. This new figure is in the supporting information, Figure S2. However, the instrument has progressed significantly since this study and now use multi-stage trapping and focusing which has greatly improved the chromatographic separation and thus rendered obsolete the performance shown here. However, we are happy to include this figure for the sake of completeness. Along with the new figure, we have added the following sentence to the main text:

"GC-Vocus chromatograms of both calibration and ambient indoor air are shown in Figure S2 to demonstrate the chromatographic separation of this system."

Section 2.8.: Did you detect any blank/background for any of the measured compounds? Degradation of Tenax TA results often to some blank (e.g benzene).

Authors' response: To check for residual backgrounds of these compounds, we ran instrument zeros (where the traps were filled with zero air during the sample collection period) and instrument blanks. In these samples, remaining peaks that were above the baseline were < 1% of the signal measured during a typical ambient sample. We appreciate this comment from the reviewer, as we have also noticed elevated signal as the trap degrades over time (e.g. benzaldehyde) and contaminations from our gases.

Table 1: Could you also give precision and uncertainty of these systems in this table or in some other part of the paper?

Authors' response: We have added 1-σ uncertainties and precisions as requested by the reviewer. As an example, the uncertainty of the sensitivity calculated from the calibration curve for α-pinene measured by the GC-Vocus was 4.9% and the accuracy of the calibration preparation was 3.0%, resulting in a total uncertainty of 5.7%. This is an explicit explanation for α-pinene; the uncertainties for the other compounds can be found in the newly added Table S1. We have also updated the text in the main paper to mention the uncertainties and precisions. The text now reads:

"From our calibration data, we estimate typical 1-σ uncertainties to be 12 % and 5 % for the GC-EI-TOF and GC-Vocus configurations, respectively, with typical precisions of 5 % and 1 %. The individual uncertainties for each calibrated compound reported from the GC are listed in Table S1."

Table 2: Even though you are able to detect some compounds (e.g. methanol, acetaldehyde etc.) I doubt their measurements are not quantitative. I would expect high breakthrough of them from the cold trap. Also some other molecules may have high losses in TD. If you have some results that show they are quantitative, please present it. If not, maybe you could mention more clearly that GC can also be used just for identification of these compounds and maybe RT-Vocus can be used for quantification of some of them?

Authors' response: The authors appreciate this comment from the reviewer, and agree that clarification in the text was needed. We have added an explanation to section 3.2 about Table 2. The text now reads:

"Table 2 reports the EI characteristic ion and the ion(s) detected by the Vocus (typically a combination of the molecular ion [MH+] along with water clusters and/or fragments) for a subset of the chromatographic peaks that were identified through the GC analysis. These identified species include alcohols, aldehydes, ketones, ethers, nitrogen containing compounds, halocarbons, siloxanes, alkanes, alkenes, and aromatics. It should be noted that not every compound listed in Table 2 can be reported quantitively from the GC system due to breakthrough in the thermal desorption trap or other losses in the system. However, even for these species that are difficult to quantify, the GC is an excellent tool for compound identification."

Line 342: Benzaldehyde has often quite high background in TD-GC runs (possibly due to degradation of Tenax TA). Even though it does not matter here, it could be more appropriate to use some other compounds as an example.

Authors' response: We agree with the reviewer that benzaldehyde can be an artifact of Tenax TA used in thermal desorption traps. However, benzaldehyde was used here as a model example of how to use the PTR mass spectrum, EI mass spectrum, and GC retention time to identify a compound. This example does not attribute the source or characterize the benzaldehyde time

profile, and so we feel it is appropriate to keep this section as is to demonstrate a "work flow" for how to use all of the information given from this system.

Section 3.4: Did you detect this compound with RT-Vocus? If so, please, show the results. This would prove that this is not coming from the TD system. Is DMSD known to have some health effects?

Authors' response: Unfortunately, in the RT-Vocus spectrum, the DMSD protonated molecular ion ($C_2H_9O_2Si+$, $m/z$ 93.0366) appears at a mass where there are large signals for species like toluene ($m/z$ 93.099), and the water clusters of $C_4H_{10}O$ ($m/z$ 93.0910) and $C_3H_6O_2$ ($m/z$ 93.0546), among others. These other species are present in the spectrum in larger amounts, causing the DMSD signal to be a minor peak on the shoulder of these others, making it difficult to quantitate.

[Figure]

With the added dimension of the GC, we are able to separate the DMSD from these other species. As noted in the text, we were concerned that the DMSD could be an artifact of the TD system so we ran humidified system zeros to check. From these experiments we found no generation of DMSD and thus ruled out its formation from the TD system itself.

The authors are not aware of DMSD health effects. However, currently very little is known about this compound.

---

## Author Comment (AC2) · 19 Nov 2020

Claflin et al. demonstrate a novel dual-channel TDPC-GC-EI/Vocus(H3O+)TOFMS instrument. The combination of chromatography with preconcentration, soft and hard ionization methods, highly time and mass resolved time-of-flight detector makes it probably the most universal and comprehensive state-of-the-art instrument currently available for time-resolved isomer-speciated VOC measurements. These measurements are particularly needed in the field of air quality and indoor and outdoor atmospheric chemistry. This manuscript shows a significant improvement in quantified chemical completeness, time resolution, and molecular speciation thanks to enormous synergy from coupling complementary state-of-the-art analytical methods. While I can see some potential for further improvement, overall, the manuscript is well written and, in my opinion, will be a valuable contribution to the fields of gas chromatography and mass spectrometry for indoor and other atmospheric applications. I would have relatively minor comments and suggestions which hopefully can be easily addressed in the revision.

Specific Comments and Suggestions
1) The title reads nicely exhaustively informative but the presence of "indoor air" in the title might be misleading. I interpret the novel instrument/method as more generally applicable than just for the indoor air but perhaps the title might mislead the AMT audience that the method/instrument is dedicated only to indoor air measurements rather than that the indoor air was just the indoor gym field example. The extremely impressive detection limit thanks to the Vocus sensitivity and preconcentration makes this method particularly powerful for discoveries also in the outdoor atmosphere and many other contexts.

Authors response: We thank the reviewer for their support of the application of this technique to many different environments (e.g. outdoor ambient applications). However, we added "isomer resolved measurements of indoor air" to the title because this technique was demonstrated for the first time during this indoor air study. While we understand the reviewer's concern about the title limiting the perceived future applications, we feel this is appropriate to leave as is, to inform the reader about the data presented in the manuscript.

2) I think the novel instrumentation presented in this paper is absolutely outstanding, but I do have a feeling that the capabilities are much greater than described in the manuscript. The table 2 nicely shows different classes but with the sub-ppt detection limit indoors one would expect thousands of ions. Are the compounds in Table 2 just select, example compounds from the weight room or was it meant to represent the complete chemical composition?

Authors response: The compounds reported in Table 2 are select examples and do not represent a complete chemical composition of the air. In general, the GC system deployed for this work was able to resolve VOCs in the volatility range of $C_5 - C_{12}$ n-alkanes, with the ability to detect non- to mid-polarity VOCs (e.g. alkanes, alkenes, aromatics, siloxanes, carbonyls). The RT-Vocus

measurements, without GC speciation, cover a wider range of VOC volatilities and the ability to detect polar compounds (e.g. carboxylic acids).

3) The paper focuses predominantly on monoterpenes, select aromatics and silicon containing VOCs (cVMS, DMSD). This is great but I would recommend expanding beyond the weight room, on the detectable compounds, ideally across a range of c*, and chemical classes. It would also be nice to add to the discussion which compounds are not detectable or are particularly challenging.

Authors response: We agree that the GC system has potential for a wide range of compounds in terms of functionality and volatility. In section 2.3 we address the capabilities and limitations of the system with the configuration used for this project:

"This column resolves non- to mid-polarity VOCs including hydrocarbons, oxygenates, and nitrogen and sulfur containing compounds, with the exception of high polarity compounds like carboxylic acids. The volatility range that the GC can resolve is a function of both the chosen GC column and the TDPC adsorbent trap. With the combination of column and adsorbent trap used for this study, the ARI GC was optimized for $C_5 – C_{12}$ hydrocarbons, along with oxygen, nitrogen, halogen, and sulfur containing VOCs."

4) I really appreciate the switching capability between the RT-Vocus, GC-Vocus and GC-EITOF. The "automatic detector switching" is emphasized already in the title. However, I could not find information how fast the switching is and how the data between switching is treated/trimmed. It would be great to include this information.

Authors response: The switching for this system is on a per chromatogram basis, not within a chromatogram. At the beginning of the GC cycle, a chromatography valve in the GC switches to the desired detector. We have updated the text in Section 2.8 to make this clearer, the text now reads:

"After the chromatographic separation, the column effluent was automatically directed to either the EI-TOF or the Vocus for detection by switching the 3rd chromatography valve (Figure 1)."

The ability to switch between detectors during a chromatogram, or to have the GC effluent split between the two detectors (like Bi et al., 2020 demonstrated) would be separate method development projects that were not a part of this work.

5) One big issue, not specific to this paper, but applicable to analytical chemistry methods in general are potential chemical conversions in the instruments or sampling system due to contact with materials (e.g. metal surface) or thermal (e.g. high temperature ramp or desorption). The authors are in an excellent position to shed some light on this question because RT-Vocus and GC-Vocus data can be directly compared for compounds which would be expected thermally unstable. I think expanding on this general issue could be interesting for the AMT community.

Authors response: We appreciate the reviewer's comment on this issue, and agree that conversions on reactive surfaces are important processes to understand (and avoid). The system

presented did implement a passivated sample flow path as described in section 2.3, however, because the system described in this work was a prototype and not a final system, we did not do an in-depth analysis of the stability of thermally labile compounds. In the future, this is a type of analysis that will be under investigation for the evolved system.

6) 100 ppq LOD for mp-xylenes is certainly extremely impressive! While 1 ppt for o-xylene is still impressive, I wonder what exactly is causing a large difference in LOD between those isomers.

Authors response: As noted in section 2.9, the LOD is a function of the standard deviation of the baseline, the full width half maximum of the chromatographic peak, and the sensitivity. The difference in LOD between the m&p- versus the o-xylene isomers is likely a function of the baseline around these chromatographic peaks. Since this work, the chromatographic peak widths have been narrowed considerably due to improvements made to the TDPC-GC system which results in lower LODs.

7) Two units are used for the normalized signals (ncps, and ncts). The text cautions the reader about the differences which helps. The normalization process is well described in Sect. 2.9. It seems that the ncps normalization was done by the second water cluster which makes me wonder if the signal was relatively constant in the Vocus at the given E/N ratio and unaffected by sample humidity. Is it assumed that this ion would reflect changes in H3O+ more than the changes in ambient H2O? Because changes in the E/N ratio would largely affect ncps values normalized to humidity-independent water cluster I would suggest adding a subscript with E/N ratio used (e.g. Sn150Td). This should allow for comparisons in future campaigns and prevent confusion of normalized sensitivities derived at different E/N ratios. I would also suggest showing in addition (maybe in parenthesis) the absolute sensitivity (cps/ppb).

Authors response: In the Vocus PTR-TOF, the water vapor added to the ion source is not pumped away separately like in a PTR-MS, and as a result the water vapor concentration in the IMR is very high and largely independent of ambient humidity. This allows the normalization of the product ion signals to a water cluster ion, which is done here because the H3O+ signal is too weak. The reviewer is correct to point out that this way of normalizing the signal depends on E/N and that it is important to explicitly mention the E/N value in the analysis. We have added "E/N = 150 Td" at the following places in the manuscript:

In section 2.9 we have updated the text to read: "Analyte signal was divided by $(H_2O)_3H+$ and multiplied by $1 \times 10^6$ cps to obtain normalized signal. This method of normalization depends on the E/N value used, which for this study was 150 Td."

Subscript on Table 1: ncps and ncts for both RT- and GC-Vocus measurements are for Vocus operation with E/N = 150 Td.

8) Further to the comment above, I have been missing some details on the Vocus operation and data processing. The reader is referred to the paper by Finewax et al. (2020) but this paper does not seem to be published yet so I could not refer to it. It is great to see the parameters for the

IMR, but it is unclear if the TPS voltages have been optimized with the Thuner or manually. I am also specifically wondering why 1.5 mbar of IMR pressure was used? It is not an issue but usually >=2 mbar is used. The E/N ratio of 150 Td is already somewhat high so the higher pressure could lower it and further boost the sensitivity but if there was a specific reason perhaps it could be interesting to include.

Authors response: An E/N of 150 Td is relatively high compared with PTR-MS instruments, but it does need to be a little higher because of the higher humidity in the IMR. This increases the amount of H3O+ relative to water dimer, allowing for more sensitive detection of aromatic compounds such as benzene, which are not ionized efficiently by protonated water dimer. Voltages were optimized using Thuner software. The IMR pressure of 1.5 mbar was chosen to limit the DC and RF voltages needed to run the IMR and avoid the risk of discharges. These experiments were done shortly after the instrument arrived in our laboratory, and more experience with operating the instrument has been gained since then. The manuscript by Finewax is close to acceptance in Indoor Air, however we have added some further details on the operation of the Vocus PTR-TOF to the text. The text for sections 2.4.2 now reads:

"The proton transfer reaction mass spectrometer used in this study is a Tofwerk Vocus PTR-TOF-MS (Tofwerk AG) described by Krechmer et al. (2018). It has nominal resolution of 12000 $m/\varDelta m$ and was operated with a resolution of 11500 at $m/z$ 150. The Vocus was operated with a data acquisition rate of 1 Hz for RT-Vocus and 5 Hz for GC-Vocus measurements. The focusing ion-molecule reactor DC and RF voltages were set to 500 V and 450 V, respectively, and was operated at a pressure of 1.5 mbar, giving a reduced electrical field (E/N) of 150 Td. Additional details of Vocus operation during this campaign are given in Finewax et al. (2020)."

9) It is nice to see the good performance of the GC-(PTR)Vocus channel. For instance, the speciating power of monoterpenes looks simply excellent. In terms of the other isomeric mixtures, would there be any benefit from using GC-(NH4+) Vocus ionization or has it not been tried yet in this configuration? Perhaps it could be inspiring to add some prognosis on this to the future work.

Authors response: Using different ionization chemistries in the Vocus (e.g. $NH_4^+$, $NO^+$) along with the GC separation is an area of current and future work, but was beyond the scope of this paper.

10) By looking at the detected compounds in the indoor campaign (Table 2) I am missing more highly oxygenated compounds such as acids, hydroxy acids. Would it be useful to try the instrument with an in-situ derivatization (e.g. Isaacman et al., 2014)? Other compound families I am wondering about detection/speciation by this GC are sulfur-containing, amides, amines, heterocycles, metalorganics. By comparing the data from the GC-Vocus and RT-Vocus, it should be possible to delineate the groups of compounds which may not have made it through the column.

Authors response: The compounds listed in Table 2 are a subset of the compounds that the GC system can resolve. As noted in our response to the reviewers comment 3, the GC system employed for this study (combination of column and traps) was able to analyze for

hydrocarbons, halocarbons, oxygenates, and nitrogen and sulfur containing compounds in the $C_5$ – $C_{12}$ n-alkane volatility range. This does include amides, amines, heterocycles, and sulfur-containing species. However, the column used (Rxi-624, Restek) is not suitable for the analysis of highly polar compounds (e.g. carboxylic acids). Currently we are working toward using our GC systems to analyze for more polar, low-volatility species through the use of different columns. And there are GC columns that are designed for the analysis of carboxylic acids; however, these columns are not suitable for the analysis of some of the compounds of interest here. So, the value of that configuration would depend on the application and science questions of the researcher.

11) It is great to see GC and Vocus synergistically complementary. I understand that Finewax et al. (2020) is going to report expanded Vocus dataset from the gym, but I wonder why D5 is shown as detected by GC-EITOF but not GC-Vocus (Table 2). It is surprising because Vocus is definitely very sensitive to D5. Could it be that the sample did not reach Vocus for some reason? What was the detectability of D6 and D7?

Authors response: D5 siloxane was unfortunately not detected by the GC-Vocus in this work due to losses in our GC-Vocus transfer line. This has been resolved since this campaign. D6 and D7 were outside the GC elution range for this system.

12) L365 The use of "artifacts" term in this context reads extremely misleading here. The fragments or clusters are typically not artifacts in PTRMS. In many cases they can be used to quantify compounds (e.g. the cyclohexadiene fragment of monoterpenes m/z 81.0699 or methanol cluster m/z 51.0446). I suggest replacing with "interferences" or "complications" to avoid confusion with artifacts from sampling tube materials, etc.

Authors response: We have updated the text as recommended by the reviewer. The sentence now reads:

"Although these interferences complicate the RT-Vocus interpretation, with the addition of GC separation the molecular identification, and identification of fragment and cluster signals, is possible to do for complex ambient samples."

13) The DMSD story is very well done. Clearly this discovery would have been much more difficult without the complementary power of this instrument. However, I am completely unconvinced by the indoor OH radical hypothesis. It simply does not make sense to me in terms of Fig 4 showing increase in concentration over night and being correlated with RH. This does seem perfectly aligned with a possibility of microbial biodegradation of siloxanes in PCPs in sweat. It would be consistent with numerous sources reporting it as a biodegradation product (Accettola et al., 2008; Xu, 1999). While this explanation seems most likely to me for this indoor air case, it does not necessarily mean that DMSD is not formed via OH oxidation outdoors which would be another example of an analogy between the atmospheric and microbial oxidation/ degradation.

Authors response: We agree with the reviewer that a potential source of DMSD is the OH oxidation of both cyclic and linear siloxanes and we acknowledge this in the text with the

following statement:

"DMSD has been shown to be an environmental degradation product of both cyclic (cVMS) and linear (PDMS) siloxanes (Rücker and Kümmerer, 2015; Tuazon et al., 2000). Both classes of organosiloxanes degrade to DMSD through gas-phase oxidation by hydroxyl (OH) radicals (Tuazon et al., 2000) and through condensed-phase hydrolysis reactions (Xu et al., 1998; Lehmann et al., 1994a,b; Lehmann et al., 1995; Carpenter et al., 1995)."

However, for the conditions of this study as described in the text (indoors with no natural light) we feel it is appropriate to rule out OH oxidation as the source of the DMSD observed in this environment. We agree with the reviewer that other formation pathways (including OH oxidation) are likely important in ambient environments. To clarify that our assessment of the role of OH oxidation is specific to this study, we have updated the text which now reads:

"For these reasons, we conclude that the DMSD observed in this study was not formed inside through gas-phase oxidation of cVMSs (e.g. $D_5$ siloxane) by OH radicals, and instead hypothesize that the production is through condensed-phase reactions followed by volatilization."

14) The quantified speciation of monoterpenes by GC-Vocus is extraordinarily skillful. These instruments are perfectly suited to contribute to a progress in source apportionments between anthropogenic, plant, fruit, and microbial sources of this important group of compounds. I strongly suspect but it would be great to know if the instrument is also capable of speciating sesquiterpenes.

Authors response: The instrument configuration presented in this paper was not set-up to resolve low-volatility compounds. However, since the work presented here, we have optimized our GC system for an expanded volatility range, including low-volatility VOCs such as sesquiterpenes, using an alternative column.

Technical
15) In several places a number and a unit are not separated by a space.

Authors response: We have gone through the manuscript and added spaces between all numbers and units.

References:
Bi, C.; Krechmer, J. E.; Frazier, G. O.; Xu, W.; Lambe, A. T.; Claflin, M. S.; Lerner, B. L.; Jayne, J. T.; Worsnop, D. R.; Canagaratna, M. R.; Isaacman-VanWertz, G. Coupling a gas chromatograph simultaneously to a flame ionization detector and chemical ionization mass spectrometer for isomer-resolved measurements of particle-phase organic compounds. Atmos. Meas. Tech. Under Review.